
# Atmosphere-Ocean-Aerosol-Chemistry-Climate Model SOCOLv4.0: description and evaluation

Timofei Sukhodolov[1,2,3], Tatiana Egorova[1,2], Andrea Stenke[2], William T. Ball[4], Christina Brodowsky[2], Gabriel Chiodo[2,5], Aryeh Feinberg[2,6,7], Marina Friedel[2], Arseniy Karagodin-Doyennel[1,2], Thomas Peter[2], Sandro Vattioni[2], and Eugene Rozanov[1,2,3]

[1] Physikalisch-Meteorologisches Observatorium Davos and World Radiation Center, Davos, Switzerland

[2] Institute for Atmospheric and Climate Science, ETH Zurich, Zurich, Switzerland

[3] St. Petersburg State University, St. Petersburg, Russia.

[4] Department of Geoscience and Remote Sensing, Faculty of Civil Engineering and Geosciences, TU Delft, Delft, the Netherlands

[5] Department of Applied Physics and Applied Mathematics, Columbia University, New York (NY), USA

[6] Institute of Biogeochemistry and Pollutant Dynamics, ETH Zurich, Zurich, Switzerland

[7] Eawag, Swiss Federal Institute of Aquatic Science and Technology, Dübendorf, Switzerland

*Correspondence to*: Timofei Sukhodolov (timofei.sukhodolov@pmodwrc.ch)

**Abstract.** This paper features the new Atmosphere-Ocean-Aerosol-Chemistry-Climate Model SOCOLv4.0 and its validation. The new model was built by interactively coupling the MPI-ESM1.2 Earth System Model (T63, L47) with the chemistry (99 species) and size-resolving (40 bins) sulfate aerosol microphysics modules from the Aerosol-Chemistry-Climate Model SOCOL-AERv2. We evaluate its performance against reanalysis products and observations of atmospheric circulation, temperature, and trace gases distribution, with a focus on stratospheric processes. We show that SOCOLv4.0 captures the low- and mid-latitude stratospheric ozone well in terms of the climatological state, variability and evolution. The model provides an accurate representation of climate change, showing a global surface warming trend consistent with observations as well as realistic cooling in the stratosphere caused by greenhouse gas emissions, although, as in previous model versions, a too fast residual circulation and exaggerated mixing in the surf zone are still present. The stratospheric sulfur budget for moderate volcanic activity is well represented by the model, albeit with slightly underestimated aerosol lifetime after major eruptions. The presence of the interactive ocean and a successful representation of recent climate and ozone layer trends make SOCOLv4.0 ideal for studies devoted to future ozone evolution and effects of greenhouse gases and ozone-destroying substances, as well as the evaluation of potential solar geoengineering measures through sulfur injections. Potential further model improvements could be to increase the vertical resolution, which is expected to allow better meridional transport in the stratosphere, as well as to update the photolysis calculation module and budget of mesospheric odd nitrogen. In summary, this paper demonstrates that SOCOLv4.0 is well suited for applications related to the stratospheric ozone and sulfate aerosol evolution, including its participation in ongoing and future model intercomparison projects.



## 1 Introduction

Global modelling of the atmosphere and its interaction with oceans, cryosphere, biosphere, and land surface dates back several decades (e.g., Manabe and Bryan, 1969). The numerical approximation of each of these Earth system components is a complex

task by itself and therefore their development often began and continued independently from each other, with simplified descriptions of missing, but important, processes included in the form of boundary conditions. With the rapid development of computational facilities and methods, the models grew in their complexity in terms of the number of processes described and the quality of their description. Motivated primarily through the context of climate change research, scientific advances in global numerical modelling have shown that, even though stand-alone approximations are still sufficient for some specific

modelling tasks, the interaction of Earth system components is required for reasonable model performance in most cases. These advances can be tracked in the history of the Climate Model Intercomparison Project requirements for models in its different phases (https://www.wcrp-climate.org/wgcm-cmip). Due to the importance of the dynamical links between the troposphere and the stratosphere (Kidston et al., 2015), many climate groups have also extended their state-of-the-art models vertically into the mesosphere (e.g., Manzini et al., 2006) or even the thermosphere (e.g., WACCM (Marsh et al., 2013) and

HAMMONIA (Schmidt et al., 2006)). For example, variations in the stratospheric polar night jet can induce significant changes in surface weather on timescales ranging from daily to long-term climate effects (e.g., Gerber et al., 2012 and references therein). Variability of this dynamical coupling can be induced by the earth system itself, i.e., responding to ocean temperature changes and vertically propagating wave-forcing from the troposphere, or by external factors such as volcanic eruptions, variations in the solar UV irradiance, and greenhouse gas changes (Kidston et al., 2015).

Middle atmosphere studies are closely linked to the representation of atmospheric chemistry since the ozone layer primarily determines the temperature structure of the stratosphere through the absorption of solar ultraviolet (UV) irradiance. This has an influence on the general circulation of the stratosphere and subsequently also on the tropospheric climate. Stratospheric ozone itself is influenced by many factors, such as the heterogeneous chemistry intensification after volcanic eruptions (e.g., Revell et al., 2016) or the acceleration of ozone destruction cycles after energetic particle precipitation events (Rozanov et al.,

2012; Mironova et al., 2015). Moreover, it is also largely affected by climate change, via radiatively-induced changes in upper stratospheric chemistry, as well as changes in the Brewer Dobson circulation (BDC, Chiodo et al., 2018). Changes in ozone can also in turn affect the BDC (e.g., Polvani et al., 2019). Therefore, changes in stratospheric ozone and dynamics feedback on each other. Ozone-circulation feedbacks have been assessed in several studies, showing their importance for stratosphere-troposphere coupling in mid-winter (Haase and Matthes, 2019; Oehrlein et al., 2020) and polar stratospheric temperature

variability in springtime (Rieder et al., 2019). Long-term stratospheric ozone variations (e.g., depletion and recovery) are able to significantly affect the tropospheric climate (Previdi and Polvani, 2014; Brönnimann et al., 2017), and even the climate response to global warming might be biased if ozone feedbacks are not taken into account (Nowack et al., 2015).

The main driving issue in middle atmosphere chemistry research was the discovery of the ozone hole in the 1980s (Farman et al., 1985). The ozone layer plays an important role in shielding the biosphere from dangerous solar ultraviolet radiation and





the risk of related increasing cases of skin cancer and other diseases induced progress in atmospheric ozone science that led to strong limitations on the production of halogen containing ozone depleting substances (hODS) in 1987 through the Montreal Protocol and its Amendments. Since then, observations and models have demonstrated the positive role of these restrictive measures (e.g. Velders et al., 2007; Egorova et al., 2013) and some signs of the ozone recovery have already been observed (Chipperfield et al., 2017). However, the expected recovery in the lower stratosphere has been questioned, based on the updated

observations (Ball et al., 2018; 2019). This issue is one of many requiring further investigation and deeper understanding. Other issues and research fields include the appearance of an unprecedentedly large ozone hole over the Northern hemisphere in spring 2020 (Witze, 2020; Manney et al., 2020); the formation of a large and deep Antarctic ozone hole in autumn 2020 (NASA Ozone Watch, https://ozonewatch.gsfc.nasa.gov/); continuous unexpected CFC-11 emissions (Fleming et al., 2020); a potential decline of the solar activity (Arsenovic et al., 2018); the potential stratospheric injection of sulfur-containing species

for solar geoengineering purposes (Tilmes et al., 2009; Vattioni et al., 2019); and a potential impact of increasing trends of iodine in the stratosphere (Koenig et al., 2020). These examples underline that our understanding of atmospheric ozone specifically, and atmospheric chemistry in general, is far from being fully resolved and inspires further model developments and studies of the ozone layer evolution, in the present and future.

The need to represent the large number of processes involved in the state evolution of the ozone layer led to the development

of atmospheric chemistry models ranging from simple box models to chemistry-transport models and finally to chemistry-climate models that include at least interactive chemistry and atmospheric dynamics, but may include ocean dynamics, aerosol microphysics and other components (https://www.sparc-climate.org/activities/ccm-initiative). The chemistry climate model SOCOL (SOlar Climate Ozone Links) was initially developed for studies related to the ozone layer (Egorova et al., 2005). Through its versions from v1 to v3 it was used with prescribed sea surface temperature and sea ice coverage fields, advancing

over time in terms of model numerics, stratospheric chemistry, and transport representation. Since the publication of the base version SOCOLv3 (Stenke et al., 2013), the atmospheric-chemistry part has undergone many further improvements, such as an addition of the volatile organic compound (VOC) chemistry, an interactive lightning NOx parameterization, corrections in schemes for solar heating rates and photolysis rates, parameterization of energetic particles, and interactive deposition schemes. The base version of Stenke et al. (2013), however, further branched into two significant sub-versions SOCOL-MPIOM with

interactive ocean (Muthers et al., 2014) and SOCOL-AER with interactive aerosol microphysics (Sheng et al., 2015), each of them receiving further, independent, upgrades (Arsenovic et al., 2018; Feinberg et al., 2019), and several smaller variants, such as an improved tropospheric ozone budget (Revell et al., 2015, Revell et al., 2018), detailed methane sources and sinks (Feinberg et al., 2018), and atmospheric selenium cycling (Feinberg et al., 2020).

The natural next step was to combine the multiple improvements and model versions into a single fourth version of the SOCOL

model by coupling these updated modules onto an upgraded dynamical core, since the core itself also underwent many improvements in recent years. As a basis for this, we used the Max Planck Institute Earth System Model (MPI-ESM1.2), so that the chemistry (MEZON) and aerosol (AER) models are attached to the atmosphere- (ECHAM6.3) ocean- (MPIOM1.6.3) land surface- (JSBACH3.2) ocean biogeochemistry- (HAMOCC6) model coupled through the OASIS3-MCT coupler. In this





paper we describe the new Atmosphere-Ocean-Aerosol-Chemistry-Climate SOCOLv4.0 model in detail (section 2) and
validate its performance against available observations and reanalysis products. The main motivation is to provide a solid
reference of model performance for future improvements and applications, including model intercomparison projects (MIPs).
The validation is split into two main parts: atmospheric dynamics (section 3.2) and atmospheric chemistry with the primary
focus on stratospheric ozone (section 3.3).

## 2 Model description

SOCOLv4.0 (SOCOLv4 hereafter) consists of the Earth System Model MPI-ESM1.2 (Mauritsen et al., 2019), the chemistry
model MEZON (Egorova et al., 2003), and the sulfate aerosol microphysical model AER (Weisenstein et al., 1997), with all
these parts being interactively coupled to each other, as schematically presented in Fig. 1. In simple terms, chemistry and
aerosol microphysics rely on atmospheric temperature, winds, and relative humidity and in turn influence the atmosphere and
ocean through the short- and long-wave radiation schemes, while aerosol microphysics depend on sulfur chemistry and provide
the aerosol surface area density and number density necessary for heterogeneous chemistry calculations. Transport of
individual gases and aerosols is performed by the flux-form semi-Lagrangian scheme of Lin and Rood (1996) in the dynamical
core of ECHAM6 that has remained unchanged from its predecessor ECHAM5. Transport is calculated every dynamical time
step (15 min). The dry and wet deposition of gases and aerosols is also based on the ECHAM6 parameters such as near-surface
turbulence and precipitation. In the following we discuss each of these main components separately and describe the latest
changes.

SOCOLv4 is based on the low resolution (LR) configuration of the MPI-ESM model. This configuration corresponds to a
spectral truncation at T63 providing an approximate horizontal grid spacing of 1.9°x1.9°. The vertical resolution of the
atmosphere is set to 47 levels from the surface to 0.01 hPa, using a hybrid sigma-pressure coordinate system. Although other
higher horizontal and vertical resolutions of MPI-ESM are also tuned and available for use, we chose the LR configuration
since it is the most used and better tuned one (Mauritsen et al., 2019), and better suited for long-term climate simulations in
terms of required computational resources and storage.

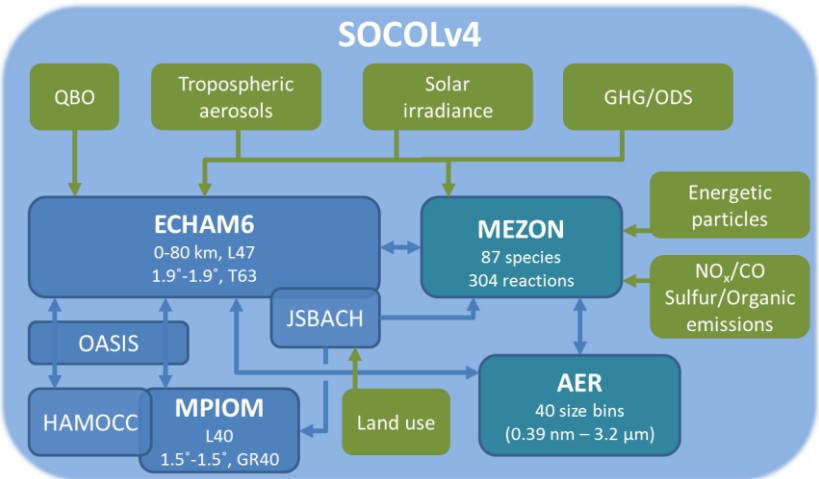

**Figure 1. Components and information flow in the atmosphere-ocean-aerosol-chemistry-climate-model SOCOLv4. Green boxes symbolize prescribed boundary conditions.**

## 2.1 Earth system model MPI-ESM1.2

The earth system model MPI-ESM1.2 (Mauritsen et al., 2019) is a further development of its predecessor MPI-ESM (Giorgetta et al., 2013). The main components of MPI-ESM are highlighted by the dark blue boxes in Fig. 1. The ocean dynamical model, MPIOM1.6.3, transports tracers of the ocean biogeochemistry model, HAMOCC6. The atmosphere model, ECHAM6.3, is directly coupled to the land model, JSBACH3.2, through surface exchange of mass, momentum, and heat. These two major model blocks are then coupled via the OASIS3-MCT coupler (Craig et al., 2017). The coupler aggregates, interpolates, and exchanges fluxes and state variables once a day between ECHAM6-JSBACH and MPIOM-HAMOCC. Here we only describe the latest states of the MPI-ESM components that are used in SOCOLv4 and do not focus on the differences between MPI-ESM versions, as this is already discussed in greater detail by Mauritsen et al. (2019). Hereafter we refer to MPI-ESM1.2 as MPI-ESM, and ignore version numberings for other components, unless otherwise stated. In terms of differences to the earlier versions, we only focus on those between the atmospheric part of the latest version, ECHAM6, and the atmospheric part used in SOCOLv3, ECHAM5.4, as changes between versions contribute the differences between the chemical response of SOCOLv4 and all sub-versions of SOCOLv3.

### 2.1.1 Ocean model MPIOM

The oceanic part MPIOM is formulated on an Arakawa-C grid in the horizontal and on z-levels in the vertical direction and solves the primitive equations with the hydrostatic and Boussinesq approximations (Jungclaus et al., 2006; 2013). Subgrid-scale parameterizations include lateral mixing on isopycnals and tracer transports by unresolved eddies. Vertical mixing is represented as a combination of the Richardson number-dependent scheme and the wind-driven turbulent mixing in the mixed layer (for details see Jungclaus et al., 2013). The horizontal grid is consistent with the MPI-ESM LR configuration, which





implies a bipolar grid GR1.5 featuring one grid pole under Greenland and one under Antarctica. The resolutions are then regionally enhanced in the deep water formation regions and the overflows across the Greenland-Scotland ridge so that the grid varies between 22 and 350 km. In the vertical, 40 levels are unevenly placed in the water column, with the first 20 levels distributed over the top 700 m. The bottom topography is represented by a partial-step formulation (Wolff et al., 1997).

The sea ice model combines the codes of MPIOM and ECHAM. In ECHAM, a simplified thermodynamic sea ice model is incorporated to provide at each atmospheric time step a physically consistent surface temperature in ice-covered regions. This part also contains a melt-pond scheme, which divides the surface of the sea ice into snow, bare ice, and melt pond with individual albedos (Pedersen et al., 2009). The atmospheric part of the sea ice model then integrates all surface fluxes into ice and provides this information to the oceanic part of the code, which uses it to calculate the sea ice surface energy balance and related changes in ice thickness. Parametrizations of changes in ice concentration and ice thickness can be adjusted by two tuning parameters, which have been used to tune the preindustrial Arctic sea ice volume of MPI-ESM to an annual average of roughly 20–25 thousands of km3 (Mauritsen et al., 2019). Sea ice dynamics is calculated following a viscoplastic approach of Hibler (1979).

### 2.1.2 Marine biogeochemistry model HAMMOC

Ocean biogeochemistry in MPI-ESM is represented by the Hamburg Ocean Carbon Cycle (HAMOCC) model (Ilyina et al., 2013; Paulsen et al., 2017). It simulates the oceanic cycles of carbon and other biogeochemical elements such as nutrients (phosphate, nitrate, and iron), oxygen, silicate, phytoplankton, zooplankton, and detritus. HAMOCC includes biogeochemical processes in the water column, the sediment, and at the air-sea interface. Biogeochemical tracers in the water column are fully advected, mixed, and diffused by the flow field of MPIOM. In total, the model has 17 state variables calculated prognostically in the water column and 12 state variables in the sediment. Nitrogen-fixing cyanobacteria was added to the model as an additional prognostic phytoplankton class by Paulsen et al. (2017).

### 2.1.3 Land surface model JSBACH

The Jena Scheme for Biosphere-Atmosphere Coupling in Hamburg (JSBACH) is the land component of MPI-ESM1.2. It provides the lower boundary conditions for the atmosphere over land and describes the dynamics of the land biogeochemistry in interaction with global climate. JSBACH treats processes like soil hydrology (five-layer scheme of Hagemann & Stacke, 2015), soil and litter decomposition, land use change (tiling approach with 12 plant functional types and two types of bare surface), fires, and a nitrogen cycle (Goll et al., 2017). Note that soil and marine chemical schemes are not yet combined with the atmospheric chemistry scheme in the current model version.

### 2.1.4 ECHAM6

ECHAM6 is an atmospheric general circulation model (GCM) that describes the large-scale circulation and its coupling to diabatic processes, both of which are ultimately driven by radiative forcing. It consists of a dry spectral-transform dynamical





core, a transport model, and a suite of physical parameterizations for the representation of diabatic processes. The prognostic variables are temperature, vorticity, divergence, logarithm of surface pressure, humidity as well as cloud ice and water. Tracer transport and diabatic processes (also referred to as "model physics") are calculated on a Gaussian transform grid. The adiabatic core of ECHAM6 consists of a mixed finite-difference/spectral discretization of the primitive equation that is identical to that employed in ECHAM5 (Stevens et al., 2013). All major changes relative to ECHAM5 are therefore related to the model

physics. These changes include: an improved representation of radiative transfer in the shortwave (or solar) part of the spectrum; a completely new description of tropospheric aerosol; an improved representation of surface albedo, including the treatment of melt ponds on sea ice (see section 2.1.1); and an improved representation of the middle atmosphere through the gravity wave forcing. In addition, minor changes have been made in the representation of convective processes. Several coding errors in model physics were also corrected in the latest version (see details in Mauritsen et al., 2019).

Transport of species is performed with the flux-form semi-Lagrangian scheme of Lin and Rood (1996). Though this scheme is mass-conservative by design, its application on the sigma-pressure coordinate system can cause a violation of the mass-conservation especially in case of large spatial gradients (Jöckel et al., 2001; Stenke et al., 2013). Turbulent mixing adopts an eddy diffusivity and viscosity approach following Brinkop and Roeckner (1995). Moist convection is parameterized according to Tiedtke (1989), with extensions by Nordeng (1994) and Möbis and Stevens (2012). Stratiform clouds are computed

diagnostically based on a relative humidity threshold (Sundqvist et al., 1989). Gravity waves are generated from a subgrid orography scheme (Lott, 1999) and as Doppler waves following Hines (1997a, b), and they are treated according to the formulation of Palmer et al. (1986) and Miller et al. (1989). Compared to ECHAM5, the tropospheric gravity wave sources are now prescribed as a function of latitude (Schmidt et al., 2013). Both long-wave and short-wave radiative transfer calculations are now described by the k-correlated method of RRTM-G (Iacono et al., 2008). Additional extra-heating

parameterization in the Hartley, Huggins, and Schumann-Runge bands and the Lyman-$\alpha$ line is now applied for better representation of the solar cycle in the mesosphere and stratosphere (Sukhodolov et al., 2014). The optical properties for radiation are updated every 2 hours. In contrast to the base model version, which applies climatological fields for this purpose, the radiation calculation of SOCOLv4 uses the prognostic tracer concentrations of sulfate aerosol and all radiatively active species (e.g., ozone and methane) except $CO_2$. Cloud scattering is parameterized according to Mie theory using maximum-

random cloud overlap and an inhomogeneity parameter to account for three-dimensional effects. Surface albedo is parameterized according to Brovkin et al. (2013). The radiative properties of tropospheric aerosols are now represented by the MACv2-SP parameterization (Stevens et al., 2017; Fiedler et al., 2017). MACv2-SP is formulated in terms of nine spatial plumes associated with different major anthropogenic source regions. It prescribes the aerosol optical depth, asymmetry factor and single scattering albedo as a function of geographical position, height above ground level, time, and wavelength. The

evolution and distribution of anthropogenic aerosol is approximated with mathematical functions, while the natural aerosol is prescribed climatologically based on Kinne et al. (2013). MACv2-SP also prescribes aerosol-cloud interactions in the form of a Twomey effect (Twomey, 1977), thus allowing the increase in the cloud droplet number concentration and associated reduction in droplet size under constant liquid water path.



## 2.2 Atmospheric chemistry model MEZON

The atmospheric chemistry part of the new model is a modified version of the chemistry-transport model MEZON (Model for Evaluation of oZONe trends; Rozanov et al., 2001; Egorova et al., 2003; Schraner et al., 2008). The last base state of this code is described in detail by Stenke et al. (2013). It underwent major upgrades for participation in the Chemistry Climate Model Initiative Phase 1 (CCMI) presented in Revell et al. (2015). Further upgrades were made in Revell et al. (2018) and Feinberg et al. (2019). Below we summarize the current state of MEZON, while the history of main updates that are now included in

SOCOLv4 is schematically illustrated in Fig. 2. In this paper, the direct comparison with SOCOLv4 is made only for the CCMI version (Revell et al., 2015) in sections 3.2.2 and 3.2.4, since it was the most widely used one in publications.

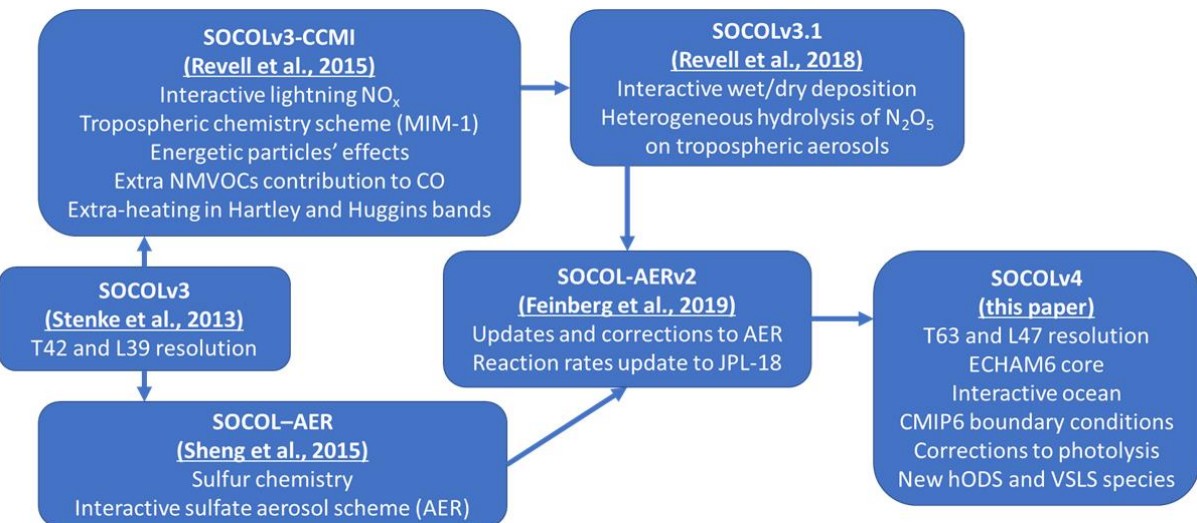

**Figure 2. History of main updates through different versions of SOCOL, which contributed to the version described in this paper.**
**Note that there were also other minor changes, adjustments, and error corrections.**

MEZON and ECHAM6 are interactively coupled by the 3-dimensional fields of temperature and wind, and by the radiative forcing induced by water vapor, ozone, methane, nitrous oxide, and chlorofluorocarbons (CFCs). The chemistry scheme is called every 2 h. The chemical solver is based on the implicit iterative Newton–Raphson scheme (Ozolin, 1992; Stott and

Harwood, 1993). The model includes 99 chemical species of the oxygen, hydrogen, nitrogen, carbon, chlorine, bromine, and sulfur groups, which are determined by 216 gas-phase reactions, 72 photolysis reactions, and 16 heterogeneous reactions in/on aqueous sulfuric acid aerosols as well as three types of polar stratospheric clouds (PSCs): supercooled ternary solution (STS) droplets, water ice, and nitric acid trihydrate (NAT). Chemical reaction rate coefficients and absorption cross sections of all reactions follow the recommendations from the NASA JPL data evaluation no. 18 (Burkholder et al., 2015). Photolysis rates

are calculated at every chemical time step using a look-up-table approach (Rozanov et al., 1999), including effects of the solar irradiance variability. Based on the stand-alone photolysis codes intercomparison study (Sukhodolov et al., 2016) and



additional sensitivity tests, the photodissociation rates of molecular oxygen and ozone (O($^1$D) path) have been supplemented by the correction factors of 1.2 for oxygen in the 10-1 hPa region and 1.5 for ozone below 100 hPa.

Dry deposition of species is based on the surface resistance approach for the estimation of dry deposition velocities proposed
by Wesely (1989). This scheme takes into account actual meteorological conditions, different surface types and trace gas properties like solubility and reactivity. Further details are given by Kerkweg et al. (2006). The wet deposition scheme is based on the SCAV scheme proposed by Tost et al. (2006). Wet deposition velocities are calculated using cloud and precipitation variables from ECHAM6, including cloud cover, liquid and ice water contents, precipitation fluxes and the convective upward mass flux. Scavenging coefficients of gaseous species depend on their Henry's law constants. Upon evaporation of clouds and
rainwater, scavenged species are transferred back into the gas phase. An additional sink for HNO3, in the form of a constant removal rate of $1\times10^{-6}$ s$^{-1}$, was introduced in the upper troposphere (above 300 hPa), compensating the missing uptake of HNO3 by ice particles (e.g., Voigt et al., 2006).

The parameterization of heterogeneous chemistry in the stratosphere is based on Carslaw et al. (1995). It coincides with HNO3 uptake by aqueous sulfuric acid aerosols resulting in the formation of STS. The parameterization of the liquid-phase reactive
uptake coefficients follows Hanson and Ravishankara (1994) and Hansen et al. (1996). The PSC scheme for water ice uses a prescribed particle number density of 0.01 cm$^{-3}$ and assumes that the cloud particles are in thermodynamic equilibrium with their gaseous environment. NAT is formed if the partial pressure of HNO3 exceeds its saturation pressure, assuming a mean particle radius of 5 μm for NAT. The particle number densities are limited by an upper boundary of $5\times10^{-4}$ cm$^{-3}$ to account for the fact that observed NAT clouds are often strongly supersaturated. The sedimentation of NAT and water ice is based on
the Stokes theory as described in Pruppacher and Klett (1997). Water ice and NAT are not explicitly transported but are evaporated back to water vapor and gaseous HNO3 after each chemical time step, transported in the vapor phase, and then depending on the saturation conditions regenerated in the next time step with the thermodynamic approximation described above. In the troposphere, the heterogeneous hydrolysis of N2O5 on tropospheric aerosols contributes to the sink of odd nitrogen. Note that since the MACv2-SP parameterization that is used for radiation does not provide the aerosol mass, for the
tropospheric chemistry we use the tropospheric aerosol climatology that considers the aerosol properties of 11 Global Aerosol Data Set types (GADS, Koepke et al., 1997) as in the previous model version. The reaction probabilities for the different aerosol types are calculated following the parametrization by Evans and Jacob (2005).

Parameterizations of the N, NO, and OH production by galactic cosmic rays, solar protons, and energetic electrons with energies of <300 keV are introduced as 0.55, 0.7, and up to 2 molecules of N, NO, and OH per ion pair respectively (Matthes
et al., 2017). The contribution of thermospheric NO from downward intrusions is parameterized as a flux-form upper boundary condition (Funke et al., 2016). The lightning source of NO$_x$ is parameterized based on the cloud top approach by Price and Rind (1992). The tropospheric budget of CO is supplemented by the Mainz Isoprene Mechanism (MIM-1, Pöschl et al., 2000) that describes the degradation of isoprene, formaldehyde and acetic acid. Contributions of all other non-methane volatile organic compounds (NMVOCs) to CO is accounted for via the addition of a certain fraction of NMVOC emissions to CO. For



anthropogenic, biomass burning, and biogenic NMVOC emissions the conversion factors to CO are 1.0, 0.31 and 0.83, respectively (Ehhalt et al., 2001).

Several newly discovered hODS (CFC-112, CFC-112a, CFC-113a, CFC114a and HCFC-133a) have been added to the model chemistry scheme together with some additional chlorine containing very short-lived substances (VSLSs: $CHCl_3$, $CH_2Cl_2$, $C_2Cl_4$, $C_2HCl_3$, $C_2H_4Cl_2$) that are not controlled by the MPA. The bromine containing VSLSs forcing is now also transient

compared to the previous model versions and includes $CH_3Br$, $CHBr_3$ and $CH_2Br_2$ species. The sulfur family is represented by eight gas-phase species: OCS, $CS_2$, $H_2S$, DMS, methanesulfonic acid (MSA), $SO_2$, $SO_3$, and $H_2SO_4$ (Sheng et al., 2015).

**2.3 Sulfate aerosol microphysics model AER**

The sulfate aerosol microphysical scheme AER is based on the two-dimensional sulfate aerosol model of Weisenstein et al. (1997). It was upgraded and combined with SOCOLv3 by Sheng et al. (2015) and later further improved by Feinberg et al.

(2019). Sulfate aerosol particles are resolved in 40 size bins, ranging in dry radius from 0.39 nm to 3.2 μm, corresponding to a range of 2.8 - $1.6\times10^{12}$ molecules of $H_2SO_4$ per particle (assuming an $H_2SO_4$ density of 1.8 g $cm^{-3}$). $H_2SO_4$ molecule number doubles between bins, while the corresponding wet sulfate aerosol radii can be much larger depending on local conditions. $H_2SO_4$ weight percent is calculated online based on actual temperature and relative humidity. The AER scheme includes submodules for the nucleation (Vehkamäki et al., 2002), composition (Tabazadeh et al., 1997), growth, evaporation (Ayers et

al., 1980; Kulmala and Laaksonen, 1990), coagulation (Fuchs, 1964; Jacobson and Seinfeld, 2004), and sedimentation of sulfate aerosol (Kasten, 1968; Walcek, 2000). In addition to gas phase $H_2SO_4$ production, the model calculates aqueous oxidation of S(IV) by ozone ($O_3$) and hydrogen peroxide ($H_2O_2$) (Jacob, 1986). The spatial distribution of cloud pH in the aqueous phase chemistry routine is approximated based on Tost et al. (2007). The aqueous production flux of S(VI) is added directly to the scavenged aerosol flux in cloud water. Dry and wet deposition of sulfate aerosol are calculated by the similar

schemes as for gas-phase species (Tost et al., 2006; Kerkweg et al., 2006). Dry and wet deposition as well as gravitational velocities of sulfate aerosol are calculated through radius-dependent parametrizations. During cloud evaporation, evaporating scavenged sulfate aerosol mass is transferred to the largest aerosol size bin. Transport of aerosols in each bin is also performed in the same way as transport of gases with a timestep of 15 min. The microphysical module is called every 2 h with 20 sub-time-steps yielding an aerosol microphysical time step of 6 min.

The influence of the aerosol on radiation fluxes at all wavelengths (14-band shortwave and 16-band longwave) is taken into account. Extinction coefficients, single-scattering albedo, and asymmetry factors required by the radiation codes are treated following a lookup table approach with precalculated aerosol physical properties using Mie theory for actual $H_2SO_4$ weight percent and temperature using refraction indices from Biermann et al. (2000). The aerosol surface area density and composition are used to calculate heterogeneous reaction rates in a chemical module.



### 2.4 Model set-up and boundary conditions


As described above, SOCOLv4 is based on the latest version of MPI-ESM (v1.2.01p6) that was also used for the Climate Model Intercomparison Project Phase 6 (CMIP6) runs. To test the new coupled model, we initialized all its parts from the MPI-ESM restart files (snapshots of the model's state at a specified time) from the end of the year 1949 and then continued the calculation until the end of 2018. The initialization of chemistry was based on SOCOLv3 runs from Revell et al. (2016).

In 1980 the reference run was split into an ensemble of three runs by imposing a temporary (1-month long) small perturbation in atmospheric $CO_2$ concentrations. For the analysis performed here, we skip the first 30 years of simulations and focus on the 1980-2018 period, which is well covered by observations for comparison with.

Model boundary conditions mostly follow the recommendations of CMIP6 provided by the input4MIPs database ([https://esgf-node.llnl.gov/search/input4mips/](https://esgf-node.llnl.gov/search/input4mips/)). All forcings are historical before 2015 and switched to the SSP2-4.5 scenario for the last

four years till 2018. These includes concentrations of greenhouse gases (Meinhausen et al., 2016), surface anthropogenic and biomass-burning emissions of $NO_x$ (as well as aircraft emissions), CO, $SO_2$, and NMVOCs (Hoesly et al., 2018), ionization rates by galactic cosmic rays, solar protons, and energetic electrons, solar spectral irradiance variations, and the influx of thermospheric of NO (Matthes et al., 2017). Biogenic emissions of all NMVOCs use a climatology for the year 2000 based on MEGAN (Model of Emissions of Gases and Aerosols from Nature; Guenther et al., 2006). For the long- and short-lived

halogenated source gases (ODSs and VSLSs) we used the WMO Ozone Assessment 2018 baseline mixing ratio scenario, which is a combined atmospheric observation record up to the year 2017 (Engel et al., 2018). For some ODSs we also used the input4MIPs database. Agricultural land-use changes are based on the Land Use Harmonization project data (LUHv2h, Hurtt et al., 2011).

Continuous degassing emissions of $SO_2$ are prescribed according to volcano locations (Andres and Kasgnoc, 1998; Dentener

et al., 2006). To represent eruptive emissions, we applied a satellite-derived dataset from Carn et al. (2016). Since the emission profiles are unknown in this database, they are emitted into the upper third of the total plume height. Air-sea DMS fluxes are calculated online through a wind-driven parametrization (Nightingale et al., 2000) and climatological gridded sea surface DMS concentrations (Lana et al., 2011). 1 Tg S yr$^{-1}$ of $CS_2$ is emitted between the latitudes of 52°S and 52°N based on Weisenstein et al. (1997). The surface mixing ratios of $H_2S$ and OCS are prescribed as 30 pptv (Weisenstein et al., 1997) and 510 pptv

(Timmreck et al., 2018), respectively.

Since the model vertical resolution is insufficient for a reasonable self-generation of the QBO, we nudge it to the observed equatorial wind profiles. The nudging is applied between 20°N and 20°S from 90 hPa up to 3 hPa. Within the QBO core domain (10°N–10°S, 50–8 hPa) the relaxation time is uniformly set to 7 days; outside this region the damping depends on latitude and altitude (Giorgetta, 1996).

Tropospheric aerosols are represented by two separate datasets for radiation and chemistry. For radiation, to be consistent with the base model and CMIP6 recommendations we used the MACv2-SP approach (Stevens et al., 2017), which however provides





only optical parameters of aerosols. Surface area densities required for the $N_2O_5$ hydrolysis, are based on the tropospheric aerosol climatology of 11 aerosol types from the Global Aerosol Data Set (GADS, Koepke et al., 1997), as in SOCOLv3.

## 3 SOCOLv4 evaluation

**3.1 Atmospheric dynamics**

### 3.1.1 Reference data for evaluation

We validate simulated climate variables from SOCOLv4 against observations. For this purpose, we use reanalysis data because they incorporate various observations combined in one data set using comprehensive techniques that allow gaps in space and time to be estimated. It should be noted that the combination of different data can lead to some errors in the reanalysis data, 335 both in trends and variability, that have a purely methodological nature, like those due to changes in the observation systems or aging of instruments as well as the details of underlying models used for assimilation. Intercomparison of different reanalysis products by Long et al. (2017) showed that reanalyses agree in the lower/middle stratosphere but deviate more from each other in the upper stratosphere and lower mesosphere because of the decreased availability of direct observations and thus the increased dependence on the assimilating model details, like the model top and vertical resolution. This should be considered 340 when making conclusions about deviations between model results and reanalysis. To validate the thermo-dynamical structure of the modelled atmosphere, we use the recent ERA5.1 reanalysis data set (Simmons et al., 2020, Hersbach et al. 2020) from ECMWF (European Center for Medium Range Weather Forecast) and MERRA-2 (Modern Era Retrospective Analysis for Research and Applications Version 2, Gelaro et al. (2017)). For surface air temperature we additionally use the BEST data (the Berkeley Earth Surface Temperatures, Cowtan et al. (2015)), which is a merged land-ocean data set.

**3.1.2 Temperature and winds**

To assess the ability of SOCOLv4 to reproduce the climate and its response to historical forcing, we show variability in annual mean surface air temperature and its anomaly (Fig. 3a, c), as well as the El Niño Southern Oscillation (ENSO) Nino3.4 index (Fig. 3b) and the Arctic sea ice extension anomaly (Fig. 3d) for the historical period 1980 to 2018. Anomalies were calculated as a deviation from the mean over the 1980-2018 period. Figure 3 (a, c) shows the global mean 2-meter temperature and its 350 anomaly simulated with the SOCOLv4 and various reanalysis data. The model shows a warming trend similar to observations, whereas the ensemble mean curve shows less inter-annual variability, as expected. As can be seen from the figure, the reanalysis data slightly disagree in terms of the absolute level and the model data are within this spread. Model surface air temperature is somewhat warmer than the ERA5.1 and MERRA-2 data, but colder than the BEST data.

ENSO is known as the largest mode of sea surface temperature anomalies which can influence atmospheric circulation (e.g., 355 Bulgin et al., 2020). Figure 3b shows variability of sea surface temperature in the Niño3.4 region (5N-5S, 120W-170W). The ERA5.1 reanalysis data (red curve) fluctuate in the Niño3.4 region in the range of ±3K. Positive fluctuations are associated





with El Niño events and negative with La Niña events. Observational data show several El Niño events with positive deviations stronger than +1.5 K in temperature on the time scale 1980-2018. with the strongest being those in 1982/1983, 1997/1998, and 2015/2016. The latest El Niño event gives the largest observed positive sea surface temperature anomalies. The SOCOLv4

simulations also show variability in the Niño3.4 region which is similar to observations but differs in phase among ensemble members. If we consider the simulated warm events, we can see that the free running model cannot reproduce the timing of the events as expected due to the chaotic nature of atmosphere-ocean coupling processes that give rise to ENSO. However, it can reproduce their amplitude, which is between -2 K and +3 K, as well as their frequency of about one event per 3-4 years. Spatial pattern of ENSO in the model is reasonable, but has some biases related to the asymmetry between El Niño and La

Niña events (for details see Tian et al. 2019). This drawback could be minimized by increasing coupling frequency between ocean and atmosphere from daily to hourly. However, this change also increases the computational demands to levels, which are less suitable for long-term climate calculations. Therefore, the LR version of MPI-ESM which we used for SOCOLv4 was tuned with a daily coupling (Mauritsen et al., 2019).

In general, SOCOLv4 accurately reproduces Arctic sea ice extent annual anomalies in line with observations, showing a similar

decline with time because of global warming. According to Ding et al. (2019), the spread in sea-ice extent between the ensemble members can be attributed to internal variability driven by fluctuations in Arctic pressure expressed as high pressure enhancing the sea ice loss and low pressure restraining it. Based on the analysis of Fig. 3d, we conclude that the model response to historical radiative forcing is sufficient and close to the observations, which proves suitability of the model for studies of past and future climate changes.



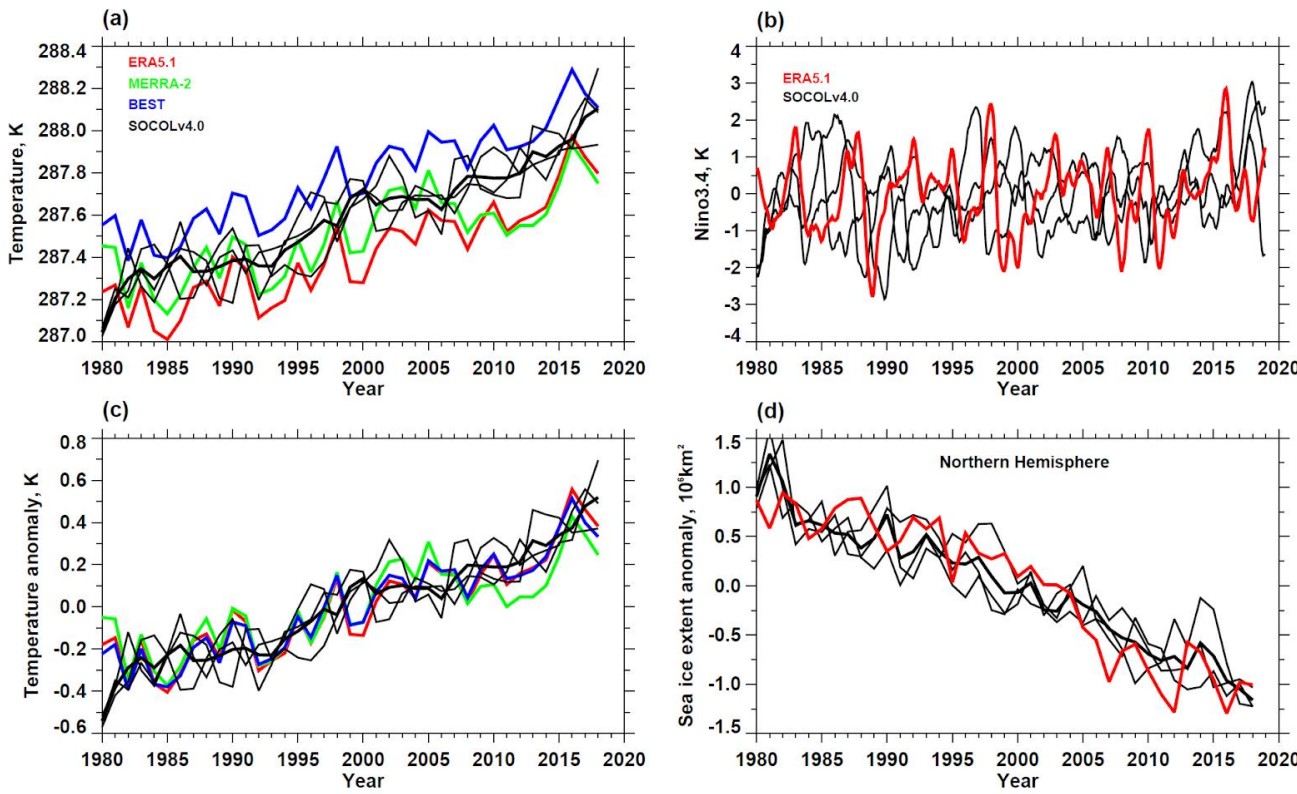


**Figure 3. Global and annual mean 2-meter temperature (a) and its anomalies relative to 1980 (c) in comparison with reanalysis data, ENSO Niño3.4 index (b), and Arctic sea ice extent annual anomaly (d). Black thick line represents simulated mean of 3-member SOCOLv4 ensemble, thin black lines are individual ensembles members, red line is ERA5.1 reanalysis, green line is MERRA-2 reanalysis, and blue line is BEST reanalysis.**


The left panel of Fig. 4 shows the geographical distribution of the 2-meter air temperature climatology for 1980-2018 as simulated with SOCOLv4 and derived from the ERA5.1 reanalysis, as well as their difference. ERA5.1 data have been interpolated to the model grid. SOCOLv4 properly reproduces the observed geographical pattern of the surface air temperature with cold high-latitude areas in both hemispheres and warm tropics and subtropics. However, there are some regional

discrepancies between the model and the reanalysis data. The model surface air temperature is generally slightly (~0.2 K) warmer than in ERA5.1, which is also seen from the global means (Fig. 3a), but there are also several spots with cold biases. In the tropical area, the discrepancies are within ±1 K, except in the eastern part of South America over the Brazilian Highlands, where negative bias reaches -5 K, and over the Tibetan Plateau and the Indochinese Peninsula, with up to -3 K bias. In the northern middle latitudes, we obtain negative discrepancies of about -5 K over the Rocky Mountains and the North Atlantic

Ocean. The most pronounced biases of ±6 K are located in the Southern Hemisphere over Antarctica. Most of these biases in high-altitude regions are a common feature of all coupled models, which is usually attributed to the simplified model's





topography and uncertainties in the observational data (e.g., Guo et al., 2020). The right panel of Fig. 4 shows the 2-m surface air temperature trends over 1980-2018 over the northern hemisphere simulated by SOCOLv4 and calculated from ERA5.1 reanalysis data. The linear trend and its significance were calculated using a robust nonparametric Sen–Mann–Kenndall test

applying 95% confidence interval. Simulated and observed trends over the Northern Hemisphere show a good spatial agreement including the location of the warming maximum north of Siberia. The maximum simulated trend amounts to 1.69±0.16 K, which is only about 0.25 K smaller than the maximum trend of 1.93 K derived from the reanalysis. Therefore, we can conclude that the climate change forcing used in SOCOLv4 and the model response to this forcing are adequately represented.

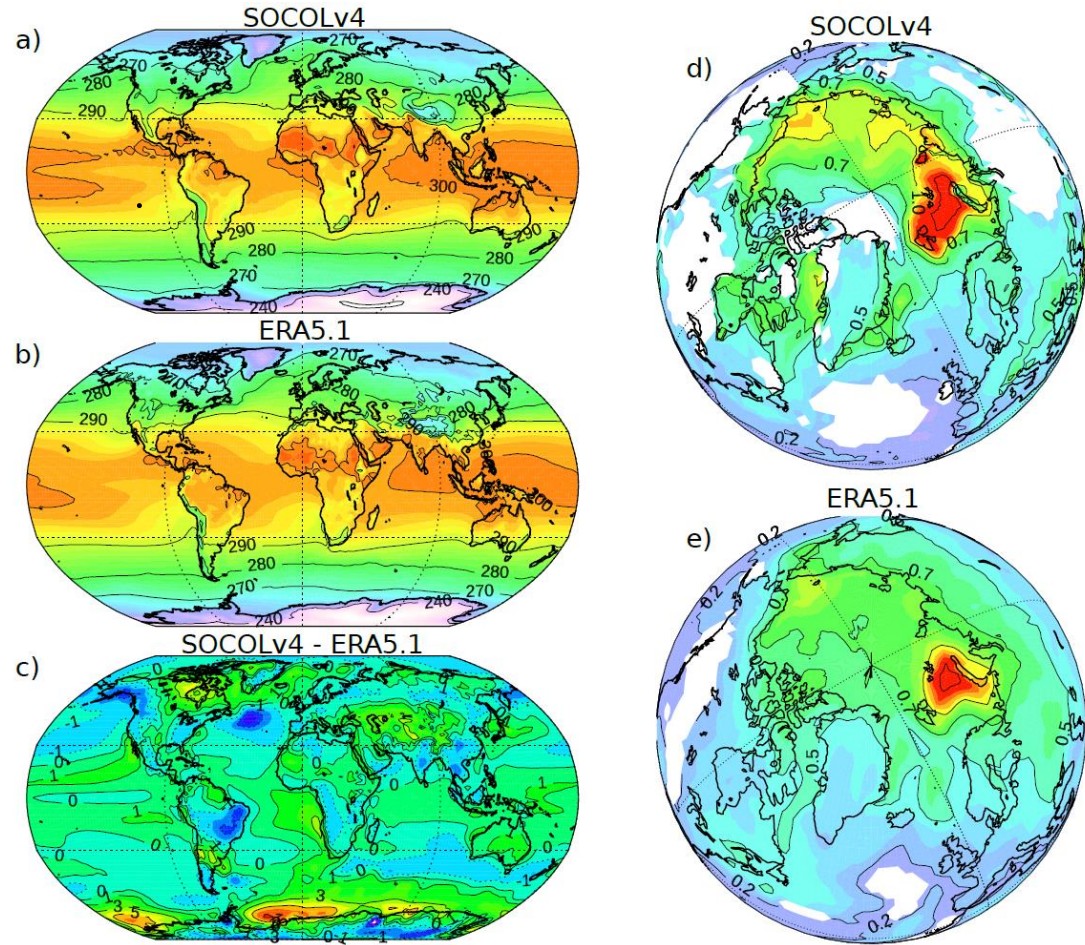


**Figure 4. Left panels: Global distribution of 2-m surface air temperature (K, contours are -5, -3, -1, 0, 1, 3, 5, 7, 9) averaged over the period 1980-2018 simulated by SOCOLv4 (a, ensemble mean), ERA5.1 reanalysis data (b), and their difference (c, SOCOL4-ERA5.1). Right panels: 2-m surface air temperature trend over the northern hemisphere (K/10years, contours are -0.5, -0.2, 0, 0.2, 0.5, 0.7, 1.0, 1.5) simulated (d) and observed (e) over the Northern Hemisphere. White spots show where the trend is not statistically**

**significant at 95% level.**



**Figure 5. Zonal mean zonal wind climatology (m/s) for DJF (left panel) and JJA (right panel) simulated with SOCOLv4 (a, b), mean of ERA5.1 and MERRA-2 reanalysis (c, d) and differences between simulated and reanalysis data (e, f). Contour intervals for the difference are -30, -20, -10, -5, -2, 2, 5, 10, 20, 30 m/s. White places denote regions where the difference between model and reanalysis data is not statistically significant at 95% confidence level calculated with the Student's t-test.**






Zonal mean climatology of zonal winds and temperatures for boreal winter and summer as simulated by SOCOLv4 are presented in comparison with observations (mean data from reanalysis ERA5.1 and MERRA-2) in Fig. 5 and 6, respectively. Even though the model has an upper limit at 0.01 hPa, we analyze the results only up to a height of 0.1 hPa, since above

ERA5.1 data are not reliable and MERRA-2 data are not available. In general, the model reproduces the observations well, but there are deviations which are, however, typical for most coupled models (Bock et al., 2020, Gettelman et al., 2019, Matthes et al., 2020). The tropospheric jets, their position and strength are reasonably reproduced by the model with biases of ±2-5 m/s. There is a tendency of the westerlies in the mid-latitudes to be insufficiently poleward, which is more pronounced in the Southern Hemisphere and represented as negative and positive deviations around 60 and 40°S, respectively. The differences

between the model and observations increase with altitude. In the stratosphere, the simulated summer-time easterly jets are of the similar strength as in observations but have some deviations in their shapes. The deviations between about 90 and 3 hPa in the (sub)tropics are related to the hemispherically symmetric QBO nudging approach, which is applied between 20°N to 20°S. The winter-time polar night jets show larger deviations in both hemispheres. In the Northern Hemisphere, the modelled polar vortex is significantly weaker than in observations and is extended equatorward in the lower mesosphere. The southern

hemispheric vortex is of the similar strength as in reanalysis mean data in the stratosphere and the main model deviations are related to its positioning. The core of the southern polar jet is shifted to 60° and located between 10 hPa and 1 hPa, while in the reanalysis data the vortex core (80 m/s isoline) extends higher and shifts towards 40°S. Therefore, the modelled southern polar vortex area is more isolated in the middle stratosphere. Oppositely, in the lower mesosphere it is extended equatorwards, similar to the northern one. The latter feature might suggest that there is some overestimation in the gravity wave forcing that

is dominant in this region (Plumb, 2002; Butchart, 2014).

The zonal mean temperature distribution is generally well reproduced by the model (Fig. 6), but there are also strong biases in the middle atmosphere with respect to the observations that are consistent with the biases in zonal winds. Namely, the winter-time high-latitude middle and upper stratosphere and the low mesosphere are warmer by 2-10 K suggesting weaker vortices and stronger meridional transport and mixing. In the rest of the stratosphere the difference is negative (about -2 K). This is

consistent with a too fast Brewer-Dobson circulation (BDC), which would cause increased adiabatic cooling in the tropics and in the summer-time high-latitudes. In the lowermost extra-tropical stratosphere, there are up to -5 K temperature deviations in both hemispheres, which leads to some downward displacement of the extratropical tropopause. The modelled tropospheric temperatures agree very well with observations and only show a warm bias over Antarctica.

It has to be noted that all of the presented model deviations in thermodynamics are long-term issues of the ECHAM-family

models and global models in general (Stevens et al., 2013; Mauritsen et al., 2019). In Figure A1, we present the same differences as in Fig. 5 and 6 but for the pure MPI-ESM CMIP6 run. As can be seen from there, SOCOLv4 and MPI-ESM share the same deviation patterns indicating weaker and warmer polar vortices in the stratosphere. Some differences between SOCOLv4 and MPI-ESM results are mostly related to the stratospheric ozone distribution at mid- and high-latitudes, as well as to the QBO nudging in SOCOLv4 in the tropics. Many of these problematic features were shown to be greatly improved





using a version of ECHAM with more detailed (95-level) vertical resolution (Schmidt et al., 2013; Stevens et al., 2013; Mauritsen et al., 2019; Matthes et al., 2020), especially the strength of the northern polar vortex and the extratropical lower-stratospheric cold biases. The increase of the vertical resolution implies, however, a strong increase in computational time, because of the column physics design. It can be partly compensated by increasing the amount of processing units, but there is also a limit in model's scalability defined by horizontal resolution.






**Figure 6. Zonal mean temperature climatology for DJF (left panel) and JJA (right panel) simulated with SOCOLv4 (a, b), mean of ERA5.1 and MERRA-2 reanalysis (c, d) and differences between simulated and reanalysis data (e, f). Contour intervals for the differences are -30, -20, -10, -5, -2, 0, 2, 5, 10, 20, 30, 40 K. White places denote regions where the difference between model and reanalysis data is not statistically significant at 95% confidence level.**





### 3.2 Atmospheric chemistry and transport

#### 3.2.1 Reference data for validation

All data sets that were used for the chemistry validation are listed in Table 1. For the comparison of HCl, $H_2O$, and $N_2O$ we used the Global OZone Chemistry And Related trace gas Data records for the Stratosphere database (GOZCARDS, Froidevaux
et al., 2015), which is a composite based on the data from 8 satellite instruments. The ENVISAT satellite Michelson Interferometer for Passive Atmospheric Sounding (MIPAS, Funke et al., 2014) instrument data was used for $NO_y$, while the SCISAT Atmospheric Chemistry Experiment Fourier Transform Spectrometer (ACE-FTS) data climatology is used for $NO_y$ and $CH_4$. Besides GOZCARDS, the ozone mixing ratio is also compared to the BAyeSian Integrated and Consolidated composite ozone (BASIC, Alsing and Ball, 2019), and the CMIP6 Ozone forcing dataset (Checa-Garcia, 2018). The latter is a
compilation of CESM1-WACCM and CMAM modelling results and is used as a standard forcing for MPI-ESM. Ozone total column is validated against Multi Sensor Reanalysis version 2 (MSRv2, Van der A et al., 2015) and SBUVv8.6 (McPeters et al., 2013) composites. Stratospheric aerosol sulfur mass is compared to MIPAS (Günther et al., 2018) and HIRS (Baran and Foot, 1994) observations and a composite recommended by CMIP6 (GloSSACv1.1, Thomason et al., 2018).

It has to be noted that observations themselves are not perfect, especially for the shorter-lived species with pronounced diurnal
cycles, low concentrations, and specific regions like high-latitudes and the upper troposphere/lower stratosphere (UTLS). Different satellite instruments vary in terms of measurement method, geographical coverage, spatial and temporal sampling and resolution, time period, and retrieval algorithm. All of this has also to be dealt with while constructing the observational composites, which imposes further potential uncertainties related to the homogenization methods applied. The SPARC Data Initiative (SPARC, 2017) overviewed the existing trace gases and aerosol measurements and provided a set of useful
recommendations concerning the related uncertainties that we used in the further sections for our validation.

**Table 1. Data used in this study for chemistry validation.**

| Data | Period | Source |
|---|---|---|
| $O_3$ mixing ratio | 1985-2012 | GOZCARDS composite (Froidevaux et al., 2015) |
| | 1985-2012 | BASICv3 composite (Alsing and Ball, 2019) |
| | 1985-2012 | CMIP6 composite (Checa-Garcia, 2018) |
| $CH_4$, NO, $NO_2$, $N_2O_5$, $HNO_3$, and $ClONO_2$ mixing ratio | 2004-2012 | ACE-FTSv3.5 (Bernath et al., 2005) |
| $H_2O$ and HCl mixing ratio | 1991-2012 | GOZCARDS |
| $N_2O$ mixing ratio | 2004-2012 | GOZCARDS |
| $NO_y$ mixing ratio | 2004-2012 | ACE-FTSv3.5 |





|  | 2002-2012 | MIPAS (Funke et al., 2014) |
| --- | --- | --- |
| $H_2O$ mixing ratio | 1991-2012 | GOZCARDS |
| Liquid $H_2SO_4$ mass | 1991-1993 | HIRS (Baran and Foot, 1994) |
|  | 1980-2016 | CMIP6 stratospheric aerosol composite (GloSSACv1.1, Thomason et al., 2018) |
|  | 2002-2012 | MIPAS (Günther et al., 2018) |
| $O_3$ total column | 1980-2018 | MSRv2 composite (Van der A et al., 2015) |
|  | 1980-2018 | SBUVv8.6 (McPeters et al., 2013) |

### 3.2.2 Trace gases climatology

In this section we discuss the SOCOLv4 performance in reproducing observed distributions of various trace species. Figures

7, 10, and 11 show the data from the model and observations and the relative differences between them. Figures 8 and 12 show

mean seasonal cycles of several species that are also compared to observations.



**Figure 7. Annually averaged zonal mean distributions of CH₄, H₂O, HCl, and N₂O (from top to bottom). First column: SOCOLv4 data. Second column: ACE-FTS and GOZCARDS observational data. Third column: relative difference between the model and observations in %. Observational missing data mask is applied to the modelling results. Note that different species have different averaging periods. White areas are either the missing data or regions where the difference between model and observations is not statistically significant at 95% confidence level.**





**CH₄**. Methane ($CH_4$) is an important source of water vapour ($H_2O$) in the stratosphere. On average two $H_2O$ molecules are
produced per $CH_4$ molecule oxidized. Methane mixing ratios drop very fast with height due to increasing availability of
hydroxyl radical (OH), the excited atomic oxygen ($O(^1D)$), and atomic chlorine (Cl). In the lower mesosphere, photolysis by
Lyman-α radiation also becomes an important sink for $CH_4$. Methane is also an important atmospheric tracer for tracking the
stratospheric circulation because of its long lifetime. $CH_4$ mixing ratio isolines in Fig. 7 are lifted upward in the low latitudes
by the ascending branch of the BDC and are pushed downward in the middle and high latitudes by the descending branch.
Isolines in the middle latitudes are flattened as a result of rapid mixing by breaking planetary waves. The balance between
horizontal mixing and the diabatic circulation produces the observed tracer slope and deviations in this distribution can hint to
potential problems in transport processes (e.g., Strahan, 2015).

SOCOLv4 nicely reproduces the methane distribution in the lower stratosphere with only about ±5% deviations from
observations. However, above ~10 hPa the model deviation gradually increases with height to up to 50% overestimation in the
lower mesosphere. The highest overestimation close to the model top in the mesosphere is likely related to the treatment of the
Lyman-α line photolysis, while the rest is due to the problems in dynamics. Thus, a too fast vertical transport in the tropics
(Fig. 7) would mean that there is less time for $CH_4$ loss on the way upwards and therefore more $CH_4$ arrives in the mesosphere,
which can be partly enhanced by underestimation of $CH_4$ photolysis. Dynamical lifetime got smaller with respect to chemical
lifetime and the tracer became more vertically mixed. The increased concentration in the upper stratosphere is then also
distributed polewards and downwards in the middle and high latitudes. Difference in skewness of the isolines suggests an
overestimation of mid-latitude mixing in the upper stratosphere/lower mesosphere, which is consistent with analysis of winds
showing that the vortices are weaker at those altitudes (Fig. 5). Middle and lower stratospheric midlatitudes, however, show
some hemispheric asymmetry with a negative bias of 5% in the SH and a positive bias of 5-10% in the NH. Figure 8 shows
the $CH_4$ seasonal cycle at 20 hPa and suggests that this anomaly is likely related to the deficiencies in horizontal mixing.
Namely, while the model reproduces very well the tropical values (1.4 ppmv isoline), the midlatitude values (1 ppmv isoline)
are excessively shifted poleward in the NH and slightly equatorward in the SH. This is also generally consistent with Fig. 5,
which shows at these altitudes a weaker NH polar vortex and the SH polar vortex that is more confined to the pole.



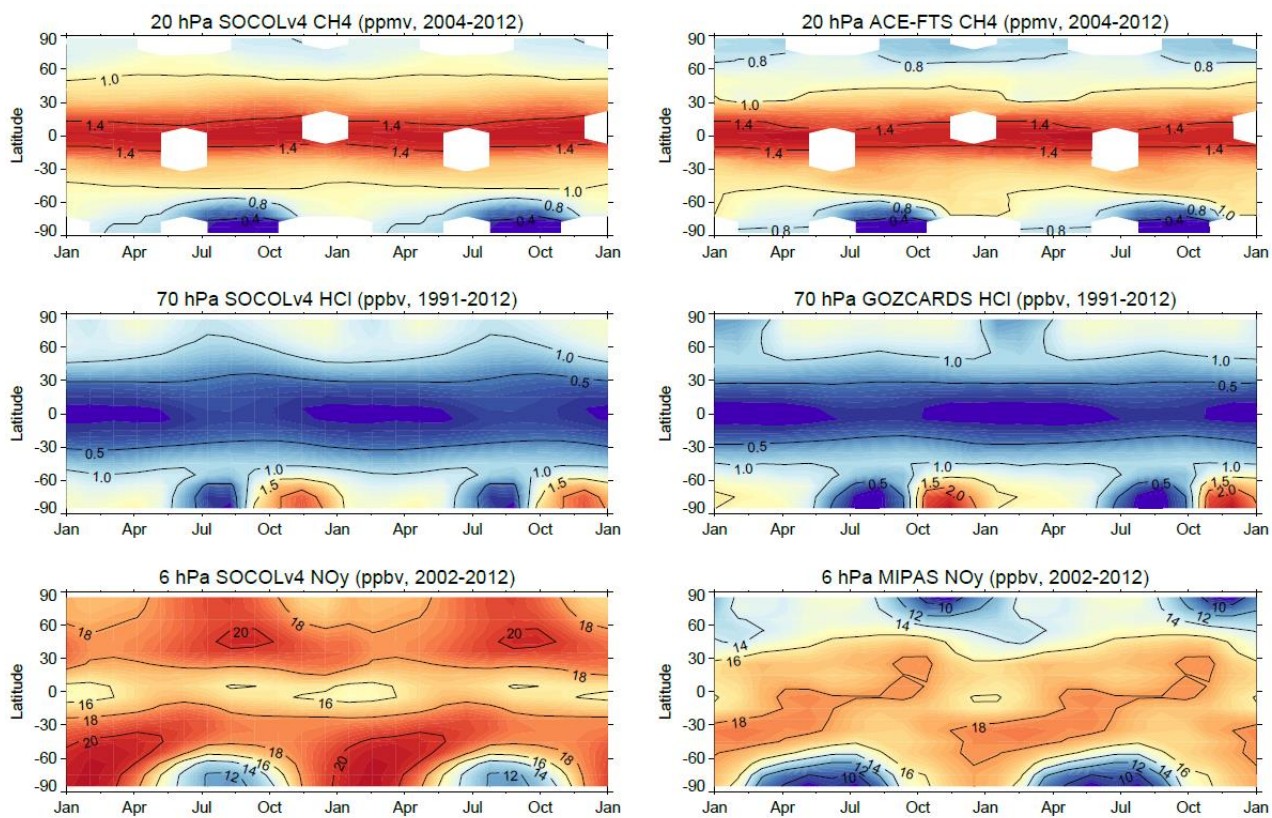

**Figure 8. CH₄, HCl, and NOᵧ zonal mean seasonal cycles at 20, 70, and 6 hPa, respectively. First column: SOCOLv4 data. Second column: ACE-FTS, GOZCARDS, and MIPAS observational data. Two consecutive cycles are shown**.

**H₂O**. Water vapour ($H_2O$) provides a source for hydroxyl radicals ($HO_x$) in the stratosphere. Hydroxyl catalytic cycles dominate the ozone destruction above the stratopause and are also very important in the lower and middle stratosphere. The two main sources of $H_2O$ in the stratosphere are methane oxidation and the transport from the $H_2O$-rich troposphere. Transport from the troposphere is important for the lower stratosphere and is highly dependent on the cold point temperature at the tropical tropopause and vertical transport in the tropics. As seen from Fig. 7, SOCOLv4 agrees very well with observations around the stratopause. In the lower stratosphere, the model is 5-10% too moist, which is due to some overestimation at the entry point. In the mesosphere, the model underestimates $H_2O$ by up to 30% (or 1 ppmv) around the model top. This partly results from the underestimated destruction of methane (about 0.2 ppmv) but mostly comes from the overestimated water vapour photolysis, which was also previously reported for SOCOL in the stand-alone photolysis codes intercomparison study (Sukhodolov et al., 2016; Karagodin-Doyennel et al., 2021). Water vapour sink (dehydration) by PSCs in the south polar lowermost stratosphere is well captured by the model.

Water vapor is a relatively long-lived species with a highly pronounced seasonality in the lower tropical stratosphere. This seasonality is related to tropospheric wave forcing, which is strongest in northern winter and weakest in northern summer,



leading to enhanced upwelling and colder temperatures at the tropical tropopause during boreal winter, and vice versa. In combination with the tropical pipe and the subtropical transport barrier, this makes $H_2O$ useful as a tracer of transport processes in the tropical stratosphere (Mote et al., 1998). Figure 9 shows the water vapor 'tape recorder' signals obtained from GOZCARDS and SOCOLv4 to illustrate the model performance in terms of vertical transport in the tropical low and middle stratosphere. SOCOLv4 reproduces the seasonal cycle of water vapor at the tropopause level well. However, SOCOLv4 shows

too fast vertical propagation of the water vapor tape recorder in the observations similar to its predecessor versions (Stenke et al., 2013), which is a common peculiarity for chemistry-climate models (e.g., Dietmüller et al., 2018). This issue is related to the problems in horizontal and vertical mixing and diffusion and also to representation of the residual circulation in general, all of which are largely dependent on resolution. The overestimation of $H_2O$ of about 0.5 ppmv throughout the middle and lower stratosphere (see also Fig. 7) is mostly due to the higher entry value during summer but can also be partly related to the

overestimation of mixing with higher altitudes.

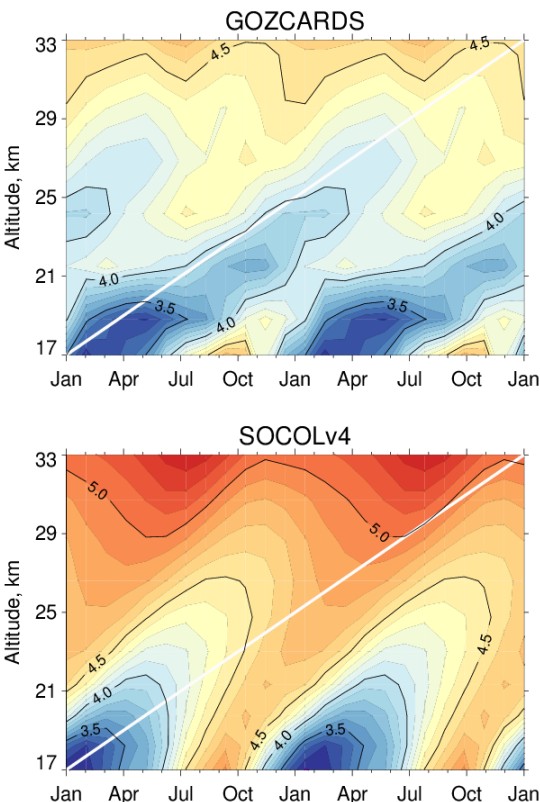

**Figure 9. Annual cycle of water vapor mixing ratio (ppmv) averaged between 15°N and 15°S and over 1991-2012 for the GOZCARDS satellite composite (upper panel) and SOCOLv4 (lower panel). Two consecutive cycles are shown. The white line in both panels indicates the approximate slope of the GOZCARDS tape recorder signal.**






**HCl**. HCl is a chlorine reservoir that is widely used for tracking the overall chlorine abundance in the stratosphere, which also plays an important role in polar ozone chemistry. SOCOLv4 reproduces the HCl climatology pretty well with deviations mostly on the order of ±5-10% throughout the stratosphere. The most pronounced deviation is the ~5% underestimation by the model at the stratopause and the tropical mid-stratosphere. Figure 8 illustrates the seasonal cycle at 70 hPa, demonstrating a rather

good representation of the southern high-latitude variations with some underestimation of the PSC activation effects, the reasons for which are yet unclear. A limiting factor here can be the amount of available $ClONO_2$ (Nakajima et al., 2016), whose observations are, however, rather sparse for these conditions. Note that HCl observational uncertainties are also the largest in the SH polar vortex (SPARC, 2017). In the northern polar regions, SOCOLv4 reproduces the semi-annual variability seen in observations, but overestimates the winter-time decrease and underestimates the summer-time one, which might be

related to the weaker northern vortex that leads to higher temperature and lower frequency of PSCs formation.

**NO$_y$**. Figure 10 shows the odd nitrogen ($NO_y=NO+NO_2+2N_2O_5+HNO_3$) climatology from the model and its difference to the ACE-FTS and MIPAS instruments data. Odd nitrogen provides another important sink of ozone in the stratosphere and also negatively contributes to and is affected by other catalytic cycles of ozone destruction by deactivating odd hydrogen, chlorine, and bromine in the reservoir species $HNO_3$, $ClONO_2$, and $BrONO_2$, respectively. The main source of $NO_y$ in the middle

atmosphere is oxidation and photolysis of nitrous oxide ($N_2O$). The lowermost stratosphere is additionally highly affected by the mixing with the troposphere and therefore the $NO_y$ budget also depends on the free tropospheric sources of odd nitrogen such as aircraft engines and lightning (Grewe, 2009). In the mesosphere, where odd nitrogen is mostly present in the form of nitric oxide (NO), the most important sources become the in-situ $N_2$ ionization by solar protons and middle energy electrons (MEE) and the transport from the thermosphere, where it is produced by auroral electrons and $N_2$ photolysis. Compared to

ACE-FTS and MIPAS, which generally agree with each other, SOCOLv4 overestimates by ~10% the maximum concentrations in the mid-upper stratosphere, the region where the odd nitrogen catalytic cycle of ozone destruction is the most effective. In the upper stratosphere and the mesosphere, the model also overestimates $NO_y$ everywhere except polar mesospheric regions. The overestimation increases gradually with height, reaching 80% in the tropical mesosphere, where the absolute values, however, are rather small. As there is almost no $N_2O$ left at these altitudes, the overestimation likely comes from the

underestimated sink through the NO photolysis. Despite the strong percentage deviation in the tropical mesosphere, the model underestimates NOy in its polar regions, though the differences are almost insignificant due to a large interannual variability. The polar bias is most likely related to an underestimated contribution of either MEE or the thermospheric NO downward flux. Mironova et al. (2019) compared CMIP6 MEE ionization effects to calculations based on polar balloon measurements and found that the related odd nitrogen signal can be up to 100% too low when CMIP6 forcing was used. High- and mid-latitude

upper stratospheric NOy concentrations are then affected by both the positive bias in the low-latitudes and the negative bias in the polar high-latitudes.

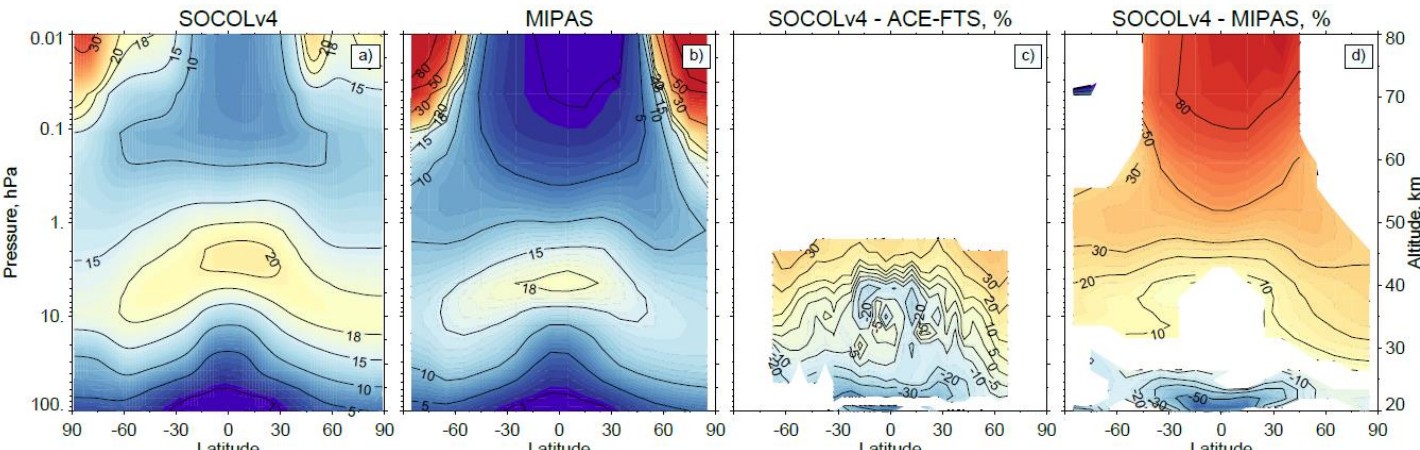

**Figure 10. NO$_y$ zonal mean annual mean climatology (ppbv) calculated by (a) the SOCOLv4 model and (b) provided by MIPAS. (c and d) - relative difference between the model and the ACE-FTS and MIPAS observations. White areas are either the missing data or regions where the difference between model and observations is not statistically significant at 95% confidence level.**

NO$_y$ observational uncertainty is the largest in the lower stratosphere (±30%, SPARC, 2017), however both instruments suggest that the model has a pronounced low bias there. This can be explained by the insufficient production from N$_2$O oxidation, which shows an underestimation in the middle and upper stratosphere (Fig. 7). This underestimation is very similar to that in methane, which suggests that both are probably related to the overestimation of the vertical transport. Another potential reason could be overestimated sinks to reservoir species, however modelled HNO$_3$ and CLONO$_2$ (Fig. A2) agree reasonably well (±10%) with observations in the areas of their maximum concentrations in the lower stratospheric mid-latitudes. The middle stratospheric overestimation of HNO$_3$ is likely related to the overestimated abundance of H$_2$O and hence HO$_x$ at these altitudes and the underestimated sink through photolysis (Sukhodolov et al., 2016). The tropical UTLS region shows large underestimation of both NO$_y$ and HNO$_3$, which suggests that either the free-tropospheric sink of odd nitrogen through HNO$_3$ washout or the source through production by lightning activity is biased. Overall, NO$_y$ representation in the model is defined by an underestimated sink in the upper stratosphere and the mesosphere, and the underestimated production in the lower stratosphere, which results in the slight vertical shift of the maximum NO$_y$ concentrations upwards.

Modelled NO$_y$ seasonal cycle at 6 hPa is presented in Fig. 8. Model shows generally higher values than MIPAS due to NO$_y$ burden being biased high, as discussed before. However, the seasonality of the variations is reasonably well reproduced at this altitude. The high and mid-latitude seasonality is controlled by the vertical motions, which either bring the NO$_y$-low air from above during the wintertime or the NO$_y$-high air from below in the summertime. This transport is modulated by horizontal mixing, which can cause some time shifts in the appearance of minima, like seen in the NH winter modelled results, where the contribution of the vertical transport is largely diminished by the mixing with midlatitudes. In the southern vortex, this is better reproduced by SOCOLv4 for the upper stratosphere, however in the middle and lower stratosphere (not shown) there are traces of the modelled southern vortex being too isolated, which is consistent with the analysis of CH$_4$ and N$_2$O. Boreal winter

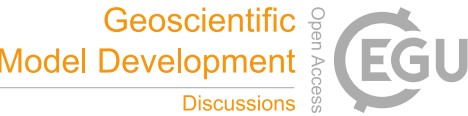

minimum in the tropics appears due to increased upwelling and, thus, transport from below, which is well catched by the model.

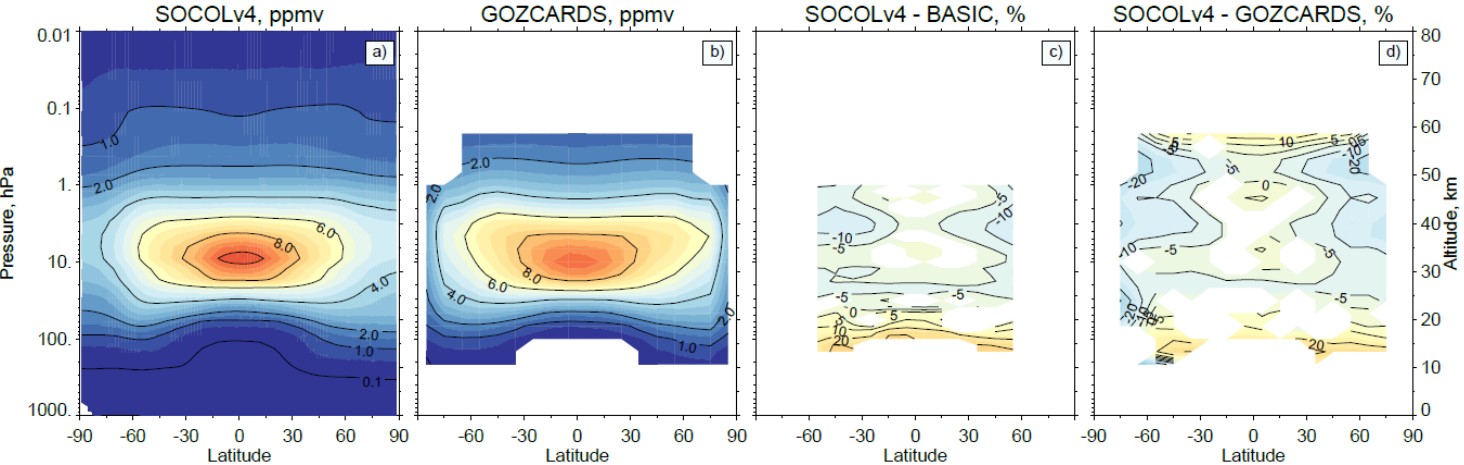

**Figure 11. Ozone zonal mean annual mean climatology (1985-2012) calculated by the SOCOLv4 model (ppmv) (a) and provided by the GOZCARDS data (b). (c and d): relative difference between the model and the observational composites BASIC and GOZCARD. White areas are either the missing data or regions where the difference between model and observations is not statistically significant at 95% confidence level.**

**O₃**. Figure 11 represents the ozone mixing ratio climatology calculated by the model versus two different ozone datasets. Both observational datasets (BASIC and GOZCARDS) agree very well with each other. The difference plots demonstrate that SOCOLv4 has a very good representation of ozone in the tropics and subtropics throughout the stratosphere, except some overestimation in its lowermost part. The lower stratospheric overestimation spreads further to mid-latitudes and reaches values of more than 20%. This overestimation is a common feature in many CMIP6 models (Keeble et al., 2020 - see their Fig.4). There is no clear chemical evidence from our previous analysis that can explain this overestimation. The low $NO_y$ model bias (Fig. 10) would also contribute negatively to the ozone burden at these altitudes through reduced production by oxidation of carbon monoxide, methane, and higher hydrocarbons, similarly to the troposphere (Gauss et al., 2006; Grewe, 2009). Therefore, it is most likely that the high ozone bias is related to the photolysis problems in SOCOLv4. Sukhodolov et al. (2016) analyzed the stand-alone performance of the look-up table photolysis scheme applied in SOCOL and found that the ozone photolysis in both channels is underestimated especially in the lower stratosphere. Although that study also showed the underestimated molecular oxygen photolysis at the same altitudes, which could partly compensate for the ozone photolysis underestimation, it appears that the latter still dominates. Comparison to the results of other codes in Sukhodolov et al. (2016) showed that the lower stratospheric problems in photolysis are probably related to the neglected Rayleigh scattering effect and temperature dependence of absorption cross sections and quantum yields. The applied photolysis corrections, as discussed in the MEZON module description, seem to improve ozone throughout the stratosphere, however a more qualitative treatment of





photolysis is already planned for the future. Insufficient ozone photolysis also might affect $HO_x$ formation via $O(^1D) + H_2O$ and related catalytic ozone destruction, thus intensifying the ozone high bias. Other factors that could contribute to the lower stratospheric overestimation in ozone are the ~2 K underestimation in local temperatures (Fig. 5), which would slow down the ozone destroying chemistry, and overestimated tropospheric ozone. In addition, SOCOLv4 so far does not include iodine
chemistry, while some recent studies suggested that it can be responsible for ~30% of the halogen-induced ozone loss in the lowermost stratosphere (e.g., Koenig et al., 2020).

In the upper stratosphere at mid-latitudes, SOCOLv4 underestimates ozone by up to -20%. This feature has been present in SOCOL models since the first version (Egorova et al., 2005) and is related to the transport issues, caused by the insufficient vertical resolution affecting the wave propagation and breaking in the stratosphere/mesosphere. Though SOCOLv4 now has
47 vertical levels compared to 39 in previous versions, all additional levels are located in the troposphere, while the number of levels in the middle atmosphere remains almost the same. As was discussed earlier, SOCOLv4 and its atmospheric base ECHAM6 show stronger residual circulation that causes stronger vertical transport in the tropics and also stronger meridional and downward transport in the winter hemisphere. The increased BDC, however, also implies a stronger wave forcing for the polar vortex making it weaker and blurring the midlatitude transport barriers (Butchart, 2014). An associated increase of
stirring, which dominates over the horizontal transport by the meridional circulation in these conditions (Plumb, 2007) would then lead to a decrease in the meridional gradient of tracers. As depicted Figure 5, the modelled polar vortex strength in both hemispheres is notably underestimated in the uppermost stratosphere and the lower mesosphere.

Figure 12 shows that the midlatitude ozone underestimation at 1 hPa appears in the winter months and that it happens together with an underestimation in the polarmost latitudes due to the more intense downward transport from the mesosphere. These
results are consistent with those seen from the analysis of methane. Part of the model biases here could be related to the 30% $NO_y$ overestimation (Fig. 10) that would intensify the ozone destruction. In the lower levels (10 hPa and 70 hPa), the model shows a generally good agreement with GOZCARDS with some deviations in timing and magnitude of extremes that can also be partly attributed to the deficiencies in transport. Thus, a too deep ozone drop at 10 hPa in the southern polar vortex is consistent with the too low methane values under these conditions and can be explained by the insufficient mixing. Seasonality
of the tropical values at all latitudes agrees very well with observations, showing a decrease of ozone in the lower stratosphere during boreal winter and a pronounced semi-annual oscillation in the middle and upper stratosphere.

Although the observations are very uncertain in the polar night regions, both the ozone mixing ratio (Fig. 12) and the total ozone measurements (Fig. 13) suggest that in SOCOLv4 the size and intensity of the Antarctic ozone hole is overestimated. This can be related to either a too strong vortex isolation, as discussed before, or to problems in the polar heterogeneous
chemistry. Since the low bias is also apparent in the higher levels, another potential reason could be the poleward dislocation of the southern night jet in the upper and middle stratosphere (Fig. 5) which would imply less upper-stratospheric ozone production inside the vortex (McConnel and Jin, 2008) and, thus, less ozone available for the downward transport. The downward transport itself might also be biased high, as illustrated by small negative anomalies of $N_2O$ and $CH_4$ in Fig. 7, which would negatively contribute to ozone by bringing too much chlorine from above.



A comparison of SOCOLv4 ozone to the prescribed CMIP6 ozone used by MPI-ESM (Fig. A3) is useful for understanding the differences in temperature and winds between SOCOLv4 and MPIESM. The comparison shows that SOCOLv4 has slightly more ozone in the tropical and subtropical upper stratosphere, but less ozone in the winter-time high latitudes. SOCOLv4 also has higher ozone in the mesosphere (up to 70%) and in the troposphere (up to 20%). Upper and mid-stratospheric differences in ozone contribute to a slightly stronger meridional temperature gradient due to less heating in the high latitudes and more

heating in the tropics. This makes the polar vortices, which are too weak in MPIESM, slightly stronger in SOCOLv4. It is pronounced the most for the SH polar vortex, which is weaker than in observations throughout the whole stratosphere in MPIESM (Fig. A3d), while in SOCOLv4 (Fig. 5f) the negative anomaly is vanished in the middle stratosphere.

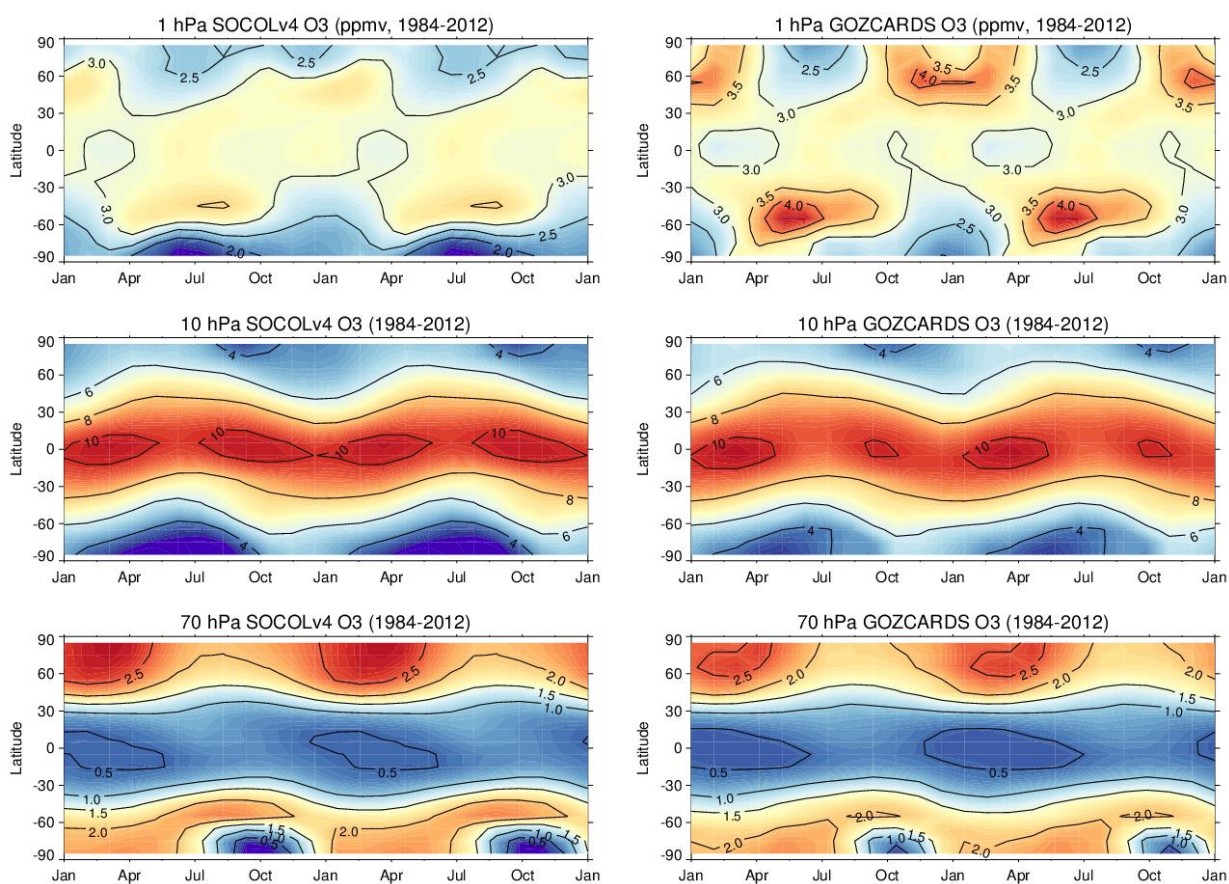

**Figure 12. 1, 10, and 70 hPa ozone zonal mean seasonal cycles (ppmv). First column: SOCOLv4 data. Second column: GOZCARDS**
**observational data. Two consecutive cycles are shown.**

### 3.2.3 Total ozone climatology

Total ozone column (TOC) represents an aggregate of all features and model biases discussed above, as well as their compensating effects. Figures 13 and 14 show that SOCOLv4 has a very good representation of tropical and subtropical TOC



compared to MSRv2 and SBUV data, which is consistent with the results of Fig. 11 and 12. The model accurately simulates
both the seasonal cycle (Fig. 13) and the annual mean climatologies (Fig. 14). Mostly due to the corrected photolysis rates,
SOCOLv4's TOC is about 20 DU lower in the tropics and subtropics than in the previous model version, which results in
better agreement with observations. Compared to MSRv2 data, SOCOLv4 shows almost perfect annual mean TOC in the
tropics with some regional underestimation of up to 5 DU. Mid-latitudes are also generally well captured, with a slight
overestimation of TOC by only 5-10 DU in both hemispheres.

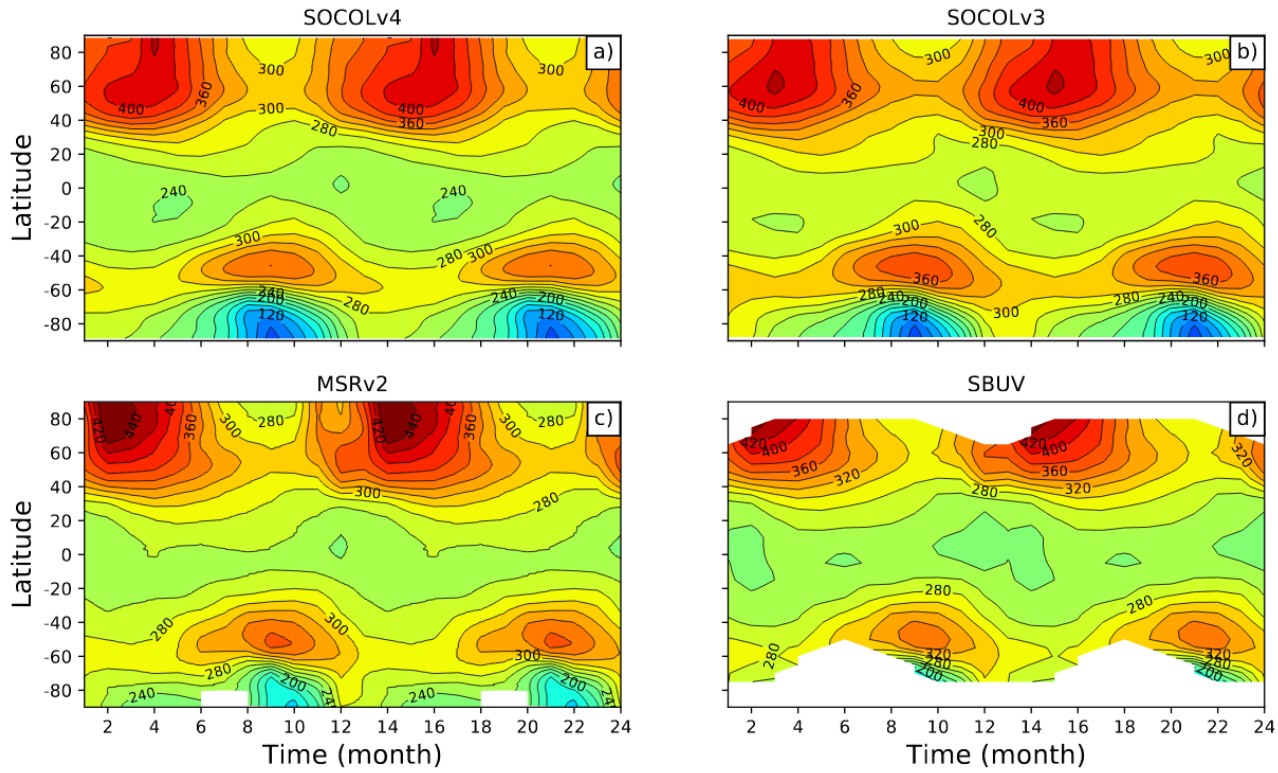


**Figure 13. Seasonal cycle of total ozone simulated by SOCOLv4 (a) against SOCOLv3 (b), MSRv2 (c), and SBUV observations (d) averaged for the 1980-2014 period.**

Major disagreement appears in polar latitudes under wintertime conditions. It has to be mentioned that the polar night
observations are subject to substantial uncertainties. SBUV data give no information for the polar night conditions at all, while
MSRv2 provides almost full spatial and temporal coverage by assimilating a large variety of data including the ground-based
observations into a chemistry-transport model (Van der A et al., 2015). This model, however, exploits a simplified scheme of
stratospheric chemistry by Cariolle and Teyssèdre (2007) that tends to overestimate TOC under the ozone hole conditions by
about 20 DU. However, taking this uncertainty into account while also using the direct satellite information by SBUV and
GOZCARDS (Fig. 11) at the edge of the polar vortices SOCOLv4 still shows some pronounced biases. Namely, the polar





ozone depletion in the SH starts right after the vortex formation (Fig. 13a), while the observations and the current state of knowledge suggests that it starts later under the spring-time conditions (Fig. 13c). This feature is fully inherited from the previous model version (Fig. 13b) and needs further detailed investigation. Likely reasons, as discussed earlier, could be related to the transport issues, the details of polar heterogeneous chemistry, or also to the treatment of photolysis under high solar zenith angles. In the Northern Hemisphere, Fig. 5 and 6 suggest a clear underestimation of the NH vortex strength in the model, which can explain the difference with respect to observations. The observations suggest some local ozone decrease in the middle of the winter, which is then followed by a fast increase in the springtime caused by inflow from the mid-latitudes. The vortex in the model is weaker, meaning that polar air masses are less isolated and can get enriched with ozone from mid-latitudes throughout the wintertime, and therefore the modelled field shows a more gradual increase instead of an abrupt rise. This stronger mixing of polar vortex air is also responsible for the NH results in Fig. 14, where the model correctly reproduces the shape of the high ozone values but also shows some overestimation in the modelled annual mean total ozone.

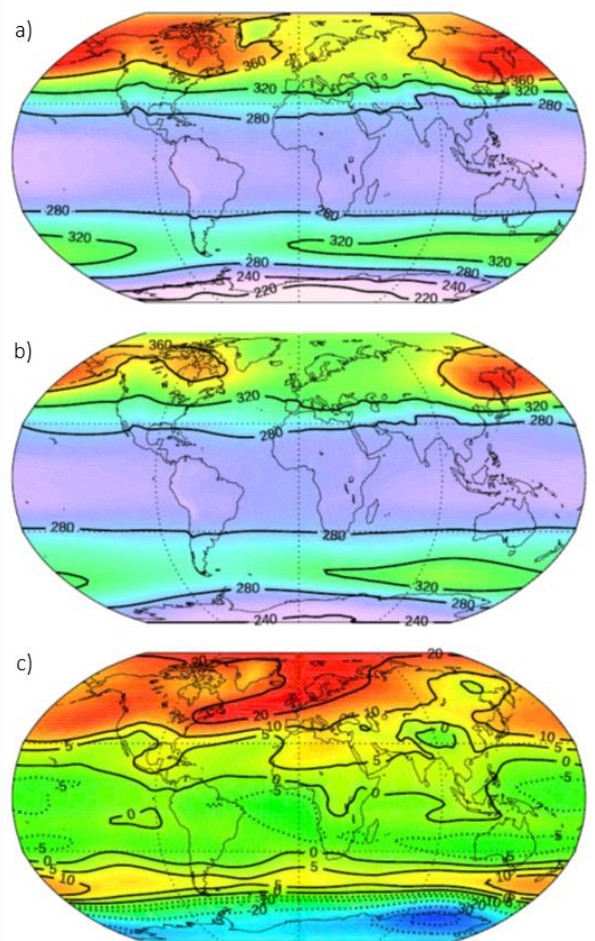

**Figure 14. Global distribution of annual mean total ozone for 1980-2018 as (a) simulated by SOCOLv4 and (b) from the MSRv2 data record in DU. (c) Difference (a)-(b) in DU with contour lines -40, -30, -20, -10, -5, 0, 5, 10, 20, 30 DU.**





### 3.2.4 Ozone evolution

The upper panel of Fig. 15 shows the evolution of the total ozone column (TOC). Near global TOC was calculated for this comparison, because SBUV has no data over polar night regions and the reliability of MSRv2 there is also low. SOCOLv3 and SOCOLv4 are presented as three separate ensemble members to illustrate the uncertainty due to the internal variability. SOCOLv3 data are taken from the RefC1 runs of CCMI-1, which assumes prescribed historical sea surface temperatures and sea ice coverages. Using prescribed SSTs in v3 explains why the variability among ensemble members is smaller in v3 than in v4, where the ocean is now interactive, and the ENSO variability is stochastic rather than fixed (Fig. 3b). It is clearly seen that SOCOLv3 has a large overestimation in TOC compared to other time-series. As was discussed above, this is mostly related to the problems in the photolysis scheme that are now manually corrected. SOCOLv4 and MSRv2 agree well in terms of the mean values, while the mean TOC from SBUV is lower by about 5-7 DU with respect to MSRv2. Generally, the slight overestimation of global TOC is also seen in many CMIP6 models, and SOCOL is well within the spread across CMIP6 models (Keeble et al., 2020, see their Fig. 6a). The CMIP6 ozone composite, instead, shows slightly lower TOC than SOCOLv4 and MSRv2 in Fig. 15a.

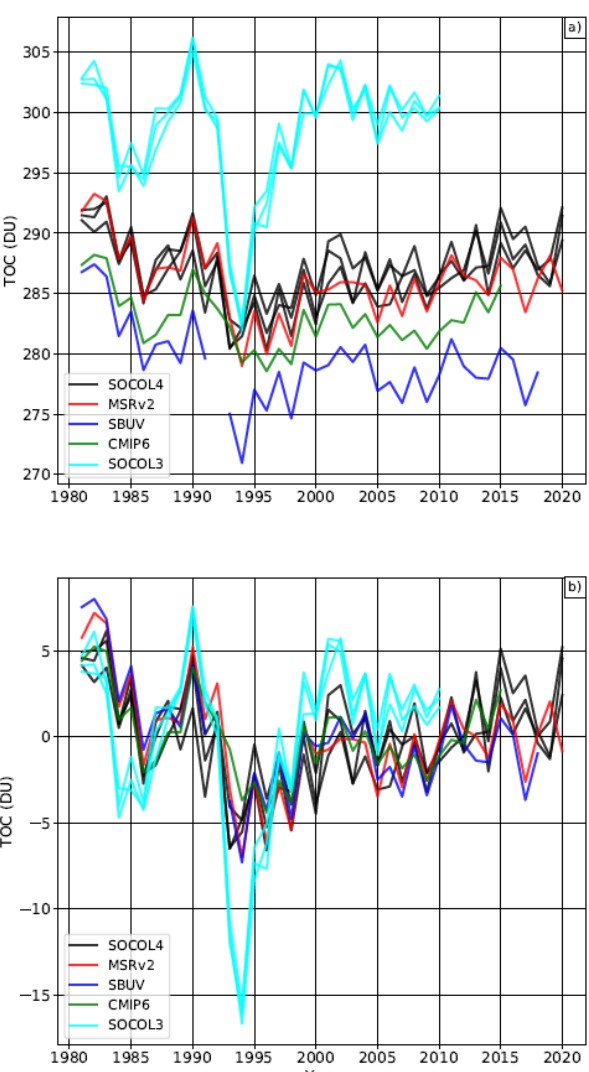

**Figure 15. Upper panel: Near global (55S-55N) mean ozone evolution simulated with SOCOLv4 (black), MSR2 (red), SBUV (dark**
**blue), SOCOLv3 (light blue), and CMIP6 (green). Lower panel: the same but normalized by the mean over the whole period.**

For easier comparison of the TOC variability we show normalized values in Fig. 15b. The four decades shown are characterized
by contributions from anthropogenic and natural forcings with varying importance during different periods. From 1980 to the
mid-90s there is a strong hODSs-induced ozone decline. This negative tendency is further enhanced by the two strong volcanic
eruptions of El-Chichon in 1982 and Pinatubo in 1991, which are then followed by several years of recovery. Since the mid-
90s, following a recovery after Pinatubo, ozone starts to increase further, induced by the leveling-out of the hODS emissions
as a consequence of the Montreal Protocol (Egorova et al., 2013; Chipperfield et al., 2015). Since the 2000s, there were no
major eruptions and the ozone evolution is mostly determined by the decreasing halogen load and the global warming effects,





though there are still many related uncertainties and other important factors in specific regions, especially in the lower

stratosphere (Ball et al., 2018; Stone et al., 2018). Upper stratospheric ozone was shown to have the most pronounced recovery (Petropavlovskikh et al., 2019; Chipperfield et al., 2018) due to being the most sensitive to the chlorine catalytic cycle of ozone destruction. Furthermore, decreasing stratospheric temperatures also led to some deceleration of ozone destruction cycles. The contribution of these positive changes to the total column is partly supplemented by an increase of the tropospheric column ozone at a rate of about 6-7%/decade resulting from the continuous increase of the surface anthropogenic emissions of ozone

precursors like CH4, CO, NOx, and VOC (Ball et al., 2018). The recovery in the middle and lower stratosphere in the tropical and extratropical areas is still unclear and observations even suggest negative trends in the lowermost stratosphere, which are, however, barely significant due to a large dynamical variability of this region (Ball et al. 2018; 2019; Petropavlovskikh et al., 2019). There are indications that these negative trends are dynamically driven (Orbe et al, 2020) and can be related to acceleration of the BDC due to rising GHGs, however models mostly fail to reproduce the observations in detail (Ball et al.,

740   2020).

In Figure 15b, we show that SOCOLv3 had a reasonable TOC evolution under volcanically quiescent conditions, while the volcanic effects were strongly overestimated. Unlike SOCOLv4, which uses an interactive sulphate aerosol scheme, SOCOLv3 used prescribed fields recommended for CCMI that seem to be biased, probably because of the uncertain observations and also the assumptions made during the calculation of aerosol mass from extinction observations. The interactive aerosol scheme

clearly improves the performance of SOCOLv4 in terms of volcanic effects on ozone. Sukhodolov et al. (2018) showed though that the observed hemispheric asymmetry of volcanic signals in ozone are still difficult to properly reproduce with a free-running model. Overall, the general behavior of the TOC evolution is very well captured by the new model both in terms of the mean values of the near-global field and the variability.

In Figure 16, we looked into some drivers of the total column evolution. Depletion and further recovery of ozone in the middle

stratosphere is well captured by the model (Fig. 16a), due to the good representation of changes in stratospheric chlorine (Figure 16e) and the continuous cooling of the stratosphere (Fig. 16b). The lower stratospheric (60 hPa) temperature and ozone observations show a much larger uncertainty, but still somewhat pronounced negative trends that are also well captured by the model. A thorough analysis of lower stratospheric processes requires comprehensive statistical tools, and a set of sensitivity runs, which is already planned for future model applications.






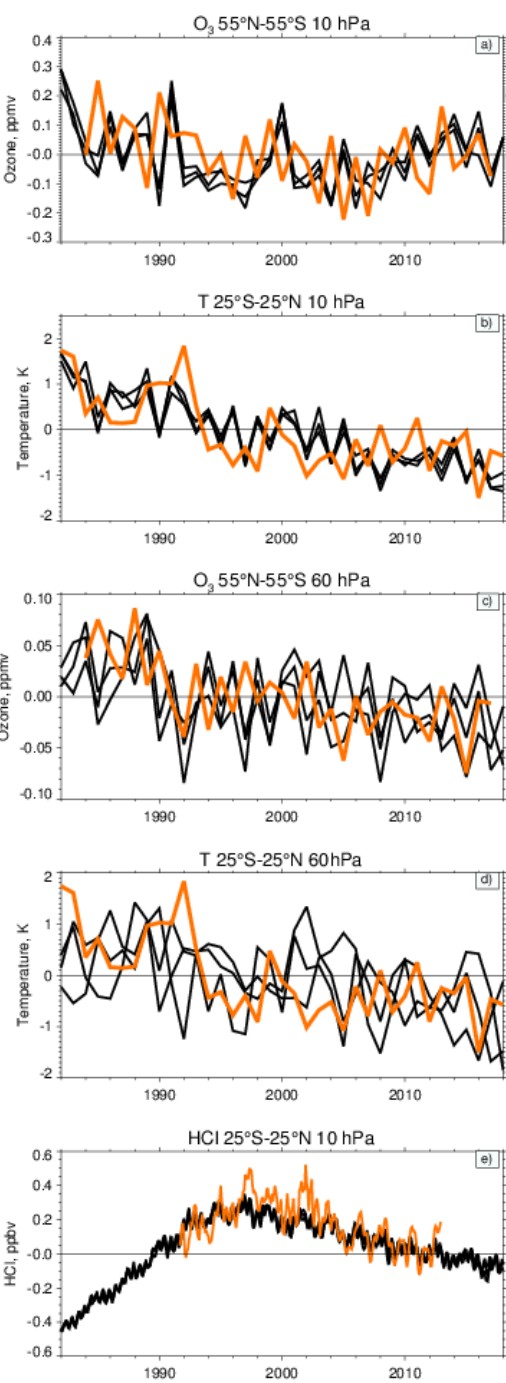

**Figure 16. (a and c): Near global (55°S-55°N) annual mean ozone evolution at 60 and 10 hPa compared to the BASIC composite. (b and d): Tropical annual mean evolution of temperature at 60 and 10 hPa compared to ERA5. (e): tropical monthly mean (smoothed with the 3-month running mean) evolution of HCl at 10 hPa compared to the GOZCARDS composite. Black lines: three model ensemble members. Orange lines: observations.**



### 3.2.5 Stratospheric sulfur

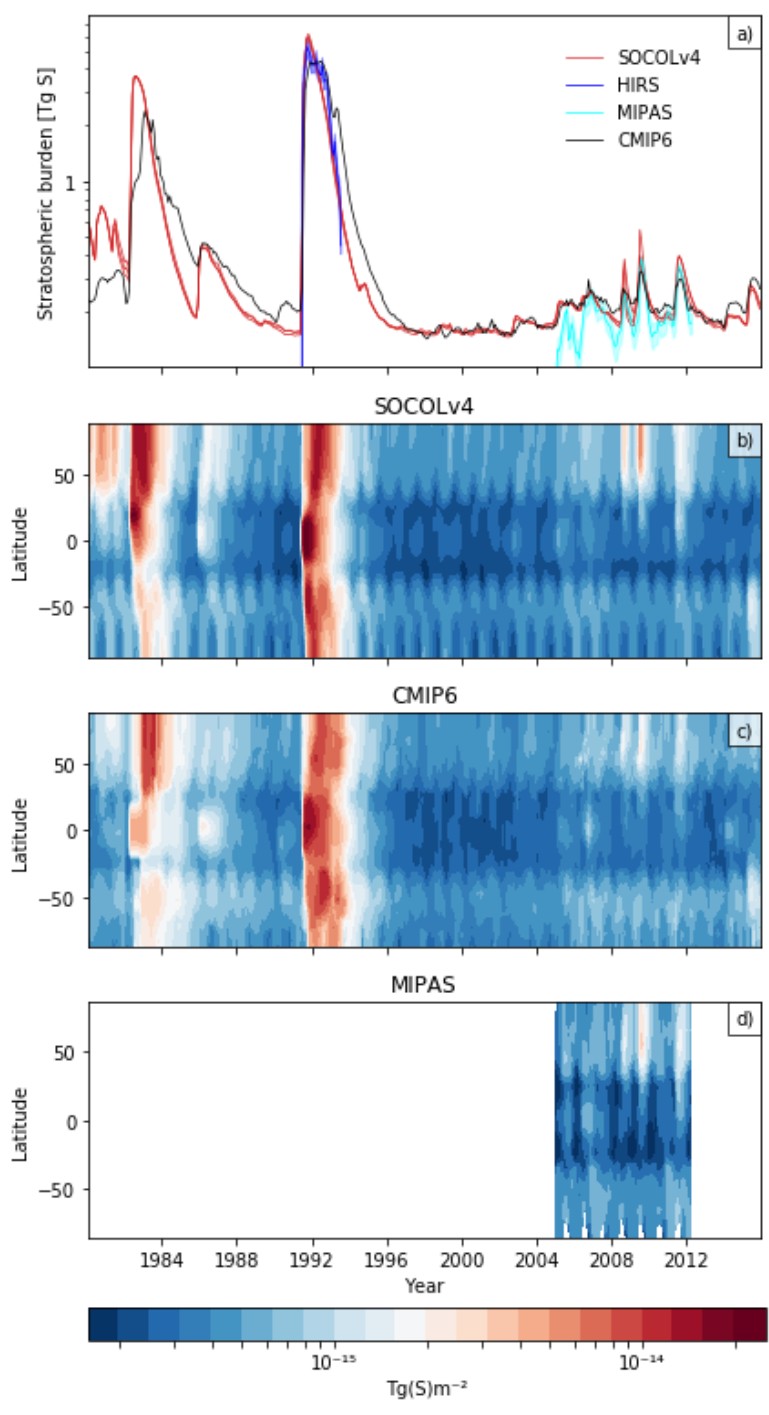

**Figure 17. Global total stratospheric sulfur mass in sulfate aerosols (a) and its zonal mean values (b-d) simulated with SOCOLv4**
**against HIRS, CMIP6, and MIPAS. Data is plotted in a logarithmic scale so that both large and small eruptions can be distinguished.**





Figure 17 shows the total stratospheric sulfur load as well as the zonal mean stratospheric column of liquid $H_2SO_4$ for SOCOLv4 against CMIP6, MIPAS and HIRS. The tropopause that was applied to calculate the stratospheric burden for the CMIP6 and MIPAS datasets was calculated from the ERA-interim temperature data using the WMO definition of the tropopause as the lowest level at which the temperature decreases less than 2 K/km, whereas for SOCOLv4 the modelled
tropopause is used. The peaks in aerosol originate from volcanic eruptions with the largest being the El Chichón (1982) and Pinatubo (1991) eruptions in the earlier time period and Kasatochi (2008), Sarychev (2009), Nabro (2011), and Calbuco (2015) as well as some others in the more recent past.

Generally, the aerosol evolution in the stratosphere is depicted quite well by the model in terms of the total aerosol amount, lifetime and meridional transport. There are, however, a few differences that need to be addressed. Prior to and after the 1982
El Chichón eruption there are some significant discrepancies between SOCOLv4 and CMIP6. In addition, the peak of stratospheric burden caused by this eruption (see Fig. 17a) is slightly shifted in time in comparison to the SOCOLv4 simulations. In Figure 17c, the delayed peak of CMIP6 is seen mostly in the northern hemisphere and the tropical aerosol levels, where the eruption took place, are lower than in SOCOLv4. The El Chichón eruption occurred between the activities of SAGE and SAGEII. Data from this time is a composite of SAMII data in the extratropics, as well several airborne lidar
missions at low latitudes, which delivered data often many months apart, leading to very sparse data coverage. The gaps were then filled by means of interpolation (Thomason et al., 2018). It is difficult to make a conclusive statement on the SOCOLv4 performance of this time due to large uncertainties in observations. The next larger event, Nevado del Ruiz in 1985, already occurred during the SAGE II era, which is deemed more reliable by Thomason et al. (2018).

The 1991 Mt. Pinatubo eruption has been discussed several times, since observations show a plateaued value for several months
after the eruption instead of a more distinct peak as models like SOCOLv4 suggest (e.g., Dhomse et al., 2020). Something similar has not been observed since then. However, no other well observed eruption was the size of the 1991 Mt. Pinatubo. A source for measurement artefacts during this time is the increase in opacity of the atmosphere due to large amounts of aerosol resulting from the eruption. From the modelling side, a potential deficiency could be insufficient vertical resolution of the stratosphere leading to increased vertical diffusion and faster meridional transport of aerosols. Niemeier and Schmidt (2017)
used a model with a similar dynamical core, ECHAM-HAM, with 39 and 90 vertical levels. They showed that the version with a higher vertical resolution maintains aerosols longer in the tropical reservoir, which increases their lifetime after low-latitude emission events. The post-plateau e-folding time looks very similar for CMIP6 and SOCOLv4, but in the CMIP6 data there is another small increase after the main Mt. Pinatubo plateau without there having been any other major eruptions. Again this could be due to a change in the instruments used for the construction of this dataset.

The volcanically quiescent period in the late 1990s and early 2000s and the minor volcanic activity later, which is also better covered by observations, are well reproduced by the model. MIPAS observations are added here, as they are not included in the CMIP6 composite and thus represent an independent source of information. MIPAS and CMIP6 agree with each other quite well in terms of elevated values after eruptions, while the background level is slightly lower in MIPAS. Background levels in SOCOLv4 are located between CMIP6 and MIPAS and there is some spread over the ensemble members, which is





caused by the variable tropopause in the model combined with vertically prescribed $SO_2$ emission profiles. The model also captures the zonal mean distribution of elevated aerosol values, reproducing some equatorward transport after high-latitude eruptions of Kasatochi, Sarychev, and Calbuco, as well as dominant northward transport after the equatorial eruption of Nabro. It should be noted that our modelling results here are largely defined by the emission database that we used (Carn et al., 2016). Other volcanic emission databases report quite different estimates for each volcanic event, both in terms of emitted amount of

$SO_2$ and its vertical distribution (Timmreck et al., 2018).

Overall, we can conclude that SOCOLv4 nicely reproduces background aerosol levels and minor volcanic activity compared to observations, while partly underestimating the aerosol lifetime after major eruptions of El Chichón, Nevado del Ruiz, and Pinatubo. The latter can be attributed to a number of factors. First of all, the data from earlier periods is less reliable and had a lower temporal coverage (Thomason et al., 2018), which influences the forcing data used in the model, as well as the

observational composites for its validation. Furthermore, the physical processes influencing aerosol concentrations after these larger eruptions differ from the smaller but better observed ones that happened more recently. Since more $SO_2$ is injected into the stratosphere at once in a larger event, OH radicals are depleted more quickly, which can cause a delay in sulfate aerosol formation. Nevertheless, all $SO_2$ is eventually (in 2-3 months time) converted to sulfate aerosol and in higher concentrations the particles will collide and grow in size. These larger aerosols are more subjected to the force of gravity and more easily

sediment to the troposphere (Timmreck et al., 2009), while the long-term decay of aerosols after smaller eruptions is more controlled by the BDC transport (Günther et al., 2018). Given that all major eruptions were tropical, a bias in vertical numerical diffusion due to insufficient vertical resolution could further intensify the sedimentation to the lower stratosphere and the troposphere and also reduce the confinement of aerosols in the tropical reservoir via increased leakage through the shallow branch of the BDC.

**4 Conclusions and outlook**

This paper presents the fourth generation of the coupled model chemistry-climate SOCOL. Unlike its predecessor chemistry-climate model SOCOLv3, SOCOLv4 includes an interactive ocean and sulfate aerosol modules. The underlying general circulation model has also been updated from ECHAM5 to ECHAM6. Our validation of the new model showed that it performs very well in terms of the mean state of most variables and the large-scale variability of the system. Namely, the warming trends

in the troposphere, the cooling trends in the stratosphere, and variability of the stratospheric and the total column ozone agree very well with observations. In the stratosphere, SOCOLv4 shows very good results for the low- and mid-latitude ozone and reasonably well reproduces climatologies of other trace gases. Compared to SOCOLv3, the new version of the model now performs significantly better both in terms of absolute values and variability of ozone, which is mostly due to corrections in the photolysis scheme and the inclusion of interactive stratospheric aerosol scheme.

However, as in the previous version, there are still some issues. Stratospheric polar night jets are weaker in the model throughout the whole stratosphere in the NH and in the upper SH stratosphere in comparison with the reanalysis data.



Accordingly, the advective Brewer-Dobson circulation and horizontal diffusion are also biased and too strong. All this affects the transport and distribution of tracers, introducing biases in $CH_4$, $N_2O$, ozone, and related species. Photochemistry itself also suffers from the outdated photolysis look-up-table (LUT) scheme, which needs to be replaced either by a LUT that also

considers temperature dependence or by an online radiative-transfer scheme. Furthermore, photolysis parameterizations of the short-wave UV need to be reviewed and updated, as we identified clear biases in the mesospheric budget of odd nitrogen, hydrogen, and oxygen. SOCOLv4 still underestimates polar winter and spring ozone in the SH, which needs further detailed investigation. Performance of the sulfate aerosol scheme is very good for the smaller eruptions and background conditions, but the duration of volcanic peaks after major events seems to be underestimated. An increase of the vertical resolution to 95

levels could be a potential source for major improvements, especially for stratospheric circulation and transport. However, it would at least double the computational needs, which might be too expensive for long-term experiments, given the number of tracers in the model. Marine emissions of species are currently prescribed in the model, but the interactive ocean and marine biogeochemistry modules now allow either to directly couple some emissions or to parameterize them as a function of other parameters. Inclusion of iodine chemistry is being currently tested and will also contribute to future updates.

Nevertheless, we showed that the model version documented here is already in a good condition to be used as a tool for studying interactions between the earth system components. The presence of the interactive ocean and a successful representation of recent climate and ozone layer trends provide a strong case for this model to be applied for studies looking at future evolution and effects of greenhouse gases and ozone-destroying substances, as well as potential geoengineering measures through sulfur injections. SOCOLv4 will also be used in the upcoming model intercomparison project CCMI phase

850 2.



**Appendix A**

**Figure A1: Zonal mean climatology differences between data simulated by MPI-ESM and the reanalysis data for temperature (a-b) and zonal wind (c-d). White places denote regions where the difference between model and reanalysis data is not statistically significant at 95% confidence level calculated with the Student's t-test.**



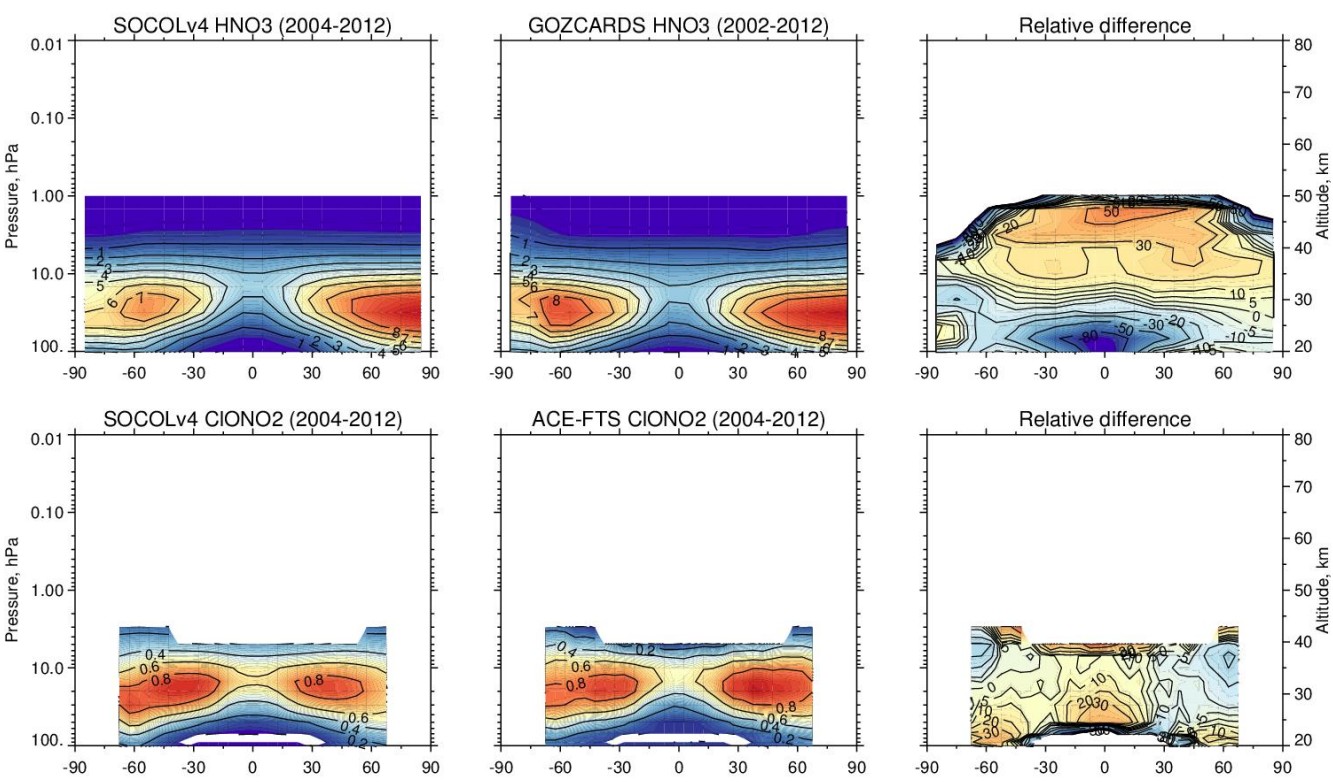

**Figure A2. HNO₃ and ClNO₂ zonal mean annual mean climatologies (ppbv). First column: SOCOLv4 data. Second column: ACE-FTS and GOZCARDS observational data. Third column: relative difference between the model and observations in %. Observational missing data mask is applied to the model data. Note that different species have different averaged periods.**

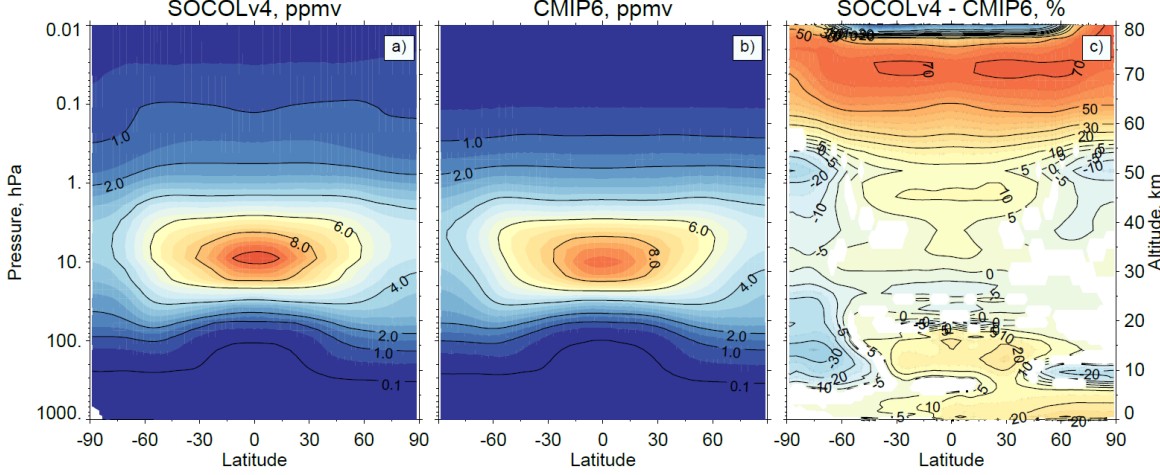

**Figure A3. Ozone zonal and annual mean climatology (1985-2012) calculated by (a) the SOCOLv4 model (ppmv) and (b) provided by the CMIP6 modelling composite. The relative difference between SOCOLv4 and CMIP6 are shown in (c). White areas are either missing data or regions where the difference between model and observations is not statistically significant at 95% confidence level.**



**Code and data availability**

The SOCOLv4.0 code can be downloaded from https://doi.org/ 10.5281/zenodo.4570622 (Sukhodolov et al., 2021). In case
of problems, please contact the corresponding author. ERA5 data is available on the Copernicus Climate Change Service (C3S)
Climate Data Store: https://cds.climate.copernicus.eu/#!/search?text=ERA5&type=dataset. MERRA-2 data are available at
MDISC, managed by the NASA Goddard Earth Sciences (GES) Data and Information Services Center (DISC):
https://disc.gsfc.nasa.gov/datasets?project=MERRA-2. The Berkeley Earth Surface Temperatures data is available at
http://berkeleyearth.org/data/. GOZCARDS data can be found here: https://gozcards.jpl.nasa.gov/info.php. The latest BASIC
ozone composite data is available at https://data.mendeley.com/datasets/2mgx2xzzpk/3. The ACE-FTS data can be found in
http://www.ace.uwaterloo.ca/climatology/3.5/netcdf/. The GloSSAC data v1.1 prepared for CMIP6 is available at
ftp://iacftp.ethz.ch/pub_read/luo/CMIP6_SAD_radForcing_v4.0.0/. MSRv2 data can be downloaded from
https://www.temis.nl/protocols/O3global.php. SBUVv8.6 merged ozone data set can be found in https://acd-
ext.gsfc.nasa.gov/Data_services/merged/instruments.html. The CMIP6 ozone composite is accessible via the input4MIPs data
server.

**Author contribution**

TS conducted the simulations, analysed the data, and drafted most of the paper. TE, AKD, and CB drafted parts of the paper.
TS, AS, AF, and ER led the model development work. All authors participated in the model development and discussions
about the results.

**Competing interests**

The authors declare that they have no conflict of interest.

**Acknowledgements**

This work was supported by the Swiss National Science Foundation (SNSF) under grants 200021_169241 (VEC) and 200020-

182239 (POLE). All calculations with the Atmosphere-Ocean-Aerosol-Chemistry-Climate Model SOCOLv4.0 were supported

by a grant from the Swiss National Supercomputing Centre (CSCS) under projects ID S-901 and ID S-1029. SNSF is also

acknowledged for support under the Ambizione Grant PZ00P2-180043 (GC and MF). SV is supported by the ETH Research

Grant ETH-17 19-2. Part of the model development was performed on the ETH Zürich cluster EULER. We thank the Center

for Climate Systems Modeling (C2SM) at ETH Zurich, the Max Planck Institute for Meteorology in Hamburg, Germany, and

personally Sylvaine Ferrachat (ETH), Urs Beyerle (ETH) and Sebastian Rast (MPI) for technical support related to hosting

and development of the model code. We also thank Bei-Ping Luo (ETH) for providing the Mie theory lookup table and Bernd

Funke (CSIC) and Michael Höpfner (KIT) for providing the MIPAS data for $NO_y$ and sulfate aerosol mass, respectively.



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
