# Peer review of "Atmosphere-Ocean-Aerosol-Chemistry-Climate Model SOCOLv4.0: description and evaluation"

_Geoscientific Model Development, 2021_

## Referee Comment (RC2)

**Review of the manuscript "Atmosphere-Ocean-Aerosol-Chemistry-Climate Model SOCOLv4.0: description and evaluation" submitted by Sukhodolov et al. for publication in GMD**

The authors describe and evaluate the new version v4.0 of the SOCOL CCM. It seems that the model performs comparatively well and I would like to congratulate the authors to this model development. I find that overall both model description and evaluation are done in an excellent way, the paper is well written, and it will be a very useful reference for future studies using this model. I have only minor suggestions that I would like to see considered before publication. I start my list with a few general issues and then proceed in the order of appearance in the text.

As a major novelty in comparison to the predecessor SOCOLv3 it is mentioned that the model is now coupled interactively to an ocean and interactive aerosol at the same time. It would be good to say clearly if the authors think that any of the evaluated features are influenced by this novelty. I wouldn't consider it a problem if not but also if that is the case it should be clearly spelled out. In this context I'm a bit at odds with the statement in the introduction that "the interaction of Earth system components is required for reasonable model performance in most cases". What are these cases? And if it really were "most cases", it should be possible to clearly identify the benefits of the full coupling in the model evaluation.

Model tuning: A major component of climate model development is the model tuning which should be adequately represented in the model description. My understanding is that the authors haven't changed anything in the model tuning in comparison to the reference model (without chemistry and aerosols) as presented by Mauritsen et al. (2019). If that is true it should be mentioned explicitly, if not, the tuning strategy would need to be described.

Computational performance: I would find some information useful on the increase of computing time due to the interactive treatment of chemistry and aerosols in comparison to the base model. I know that this may be machine and configuration dependent but some numbers for typical configurations would be informative.

Figures: Most of the figures appear fairly blurred in the pdf accessible to me. This should be taken care of before publication.

L95: "an upgraded dynamical core" Has there really been a change of the dynamical core in the family of ECHAM atmosphere GCMs since version 4? If yes, please specify.

L153 My understanding is that the sea ice model has plenty more than just two parameters that could be used for tuning. But it is also not clear to me what the

authors mean when they speak of "parameterizations of changes in ice". As said above, if the authors did any tuning it should be mentioned.

L193 "tropospheric GW sources prescribed as a function of latitude": Which GWs? Was this option used in these experiments? If yes, using what function.

L194 "RRTMG": If SOCOL follows Mauritsen et al. (2019), then the PSRAD variant of RRTMG is used, which should be mentioned.

L198 "radiation calculation of SCOCOLv4": I guess this is true only for the radiation calculation interacting with the model dynamics, not photochemistry, right? It would be useful to mention that these things are, to my understanding treated inconsistently as it is done in many other CCMs.

L198 "uses prognostic tracer concentrations … except CO2" This seems to clash with L305 where it is claimed that concentrations of GHGs from scenarios are used. Or do you mean emissions for the latter? What about $CH_4$ and $N_2O$, which are not mentioned at line 305?

L202 MACv2-SP: I thought that SOCOL calculates at least sulfate aerosols interactively. Please explain where the climatology is used and where interactive aerosol.

3.1.2: In particular here it would be interesting to know if SOCOL performs in any way different than the base model without interactive chemistry and aerosols.

ENSO evaluation: I don't think a Nino3.4 time series is very useful for evaluating the model performance. A power spectrum and teleconnection patterns would be much more helpful. But again, if SOCOL performs very similar to the base model it would be sufficient to just state that.

L398 "conclude that forcing and model response are adequately represented": Well in principle it could also be that the forcing is too strong and the response too weak.

L422 It would be good to explain the QBO nudging approach or to provide a reference.

L615: For the ozone bias in the lowermost stratosphere only chemistry is discussed as a potential reason. Could it be that the tropopause is at a slightly wrong altitude which may have large effects in this regions of large vertical gradients?

L656 "southern (polar) night jet"

Figure in the appendix: I don't know the journal policy concerning this, but if the appendix contains only three figures I would consider it more convenient to not have an appendix but include the figures in the main part.

Figure 15: I wouldn't speak of "normalized", but that an anomaly with respect to some reference is presented.

L769 "whereas for SOCOlv4": I find the whereas confusing because at the first read I related it to the tropopause definition.

L807 "underestimating the aerosol lifetime" Well, if the observations are flawed, as mentioned in the following sentence, it's not sure that this is an underestimation.

L822 "SOCOL includes an interactive ocean" As said in the beginning: If this is put so prominently as an achievement, effects of that should be discussed.

---

## Author Comment (AC1)

**Authors' response to the Referees' comments on "Atmosphere-Ocean-Aerosol-Chemistry-Climate Model SOCOLv4.0: description and evaluation"**

We thank both referees for their constructive comments and thorough evaluation of the manuscript, which helped us to improve it and to make certain parts clearer. Following the comments of both referees, we have performed an additional analysis of the ENSO data.

Referees' comments are repeated below in blue and our response follows in black. Parts of the manuscript text are shown in italic and the corrected text is highlighted in bold.

**Response to comments by Referee 1**

Page 3, lines 96-98: This is very minor - you introduce the component models and their acronyms here, but the references aren't given until Section 2. Objectively, this is fine as everything is referenced, but as a new reader I noticed that no references were given.

We decided not to overwhelm the text with repeating references, as they already appear few lines later in the beginning of Section 2. Instead, we have added some words in bold: "*In this paper we describe the new Atmosphere-Ocean-Aerosol-Chemistry-Climate SOCOLv4.0 model **and its components** in detail (section 2)…*", so that readers get a hint that more details are to follow soon.

Figure 1: The colours used for the blue and turquoise boxes are quite similar, could you have a greater colour contrast?

We have changed the colour for a greater contrast. We've also found a mistake regarding the number of MEZON species: it must read 87 instead of 99.

Figure 1: Tropospheric aerosols are listed as a boundary condition for both ECHAM6 & MEZON, and I was wondering how these were different from the aerosols calculated from AER? Are these BCs purely the SADs for N2O5 hydrolysis?

AER sulfate aerosols are used in the stratosphere only. In the troposphere, we have two sources of tropospheric aerosols, which also include sulfate: MACv2-SP (optical properties) for radiation and GADS (ND and SAD) for chemistry. We have corrected several parts throughout the text to make this clearer. In SOCOLv3 we used GADS for both the radiation and chemistry, but in SOCOLv4 we decided to use MACv2-SP to be "radiatively" consistent with the new base model MPI-ESM. However, MACv2-SP doesn't provide data about the mass of aerosols, so we still had to rely on GADS data for N2O5 hydrolysis calculations.

On page 5, lines 133-134 you state "Hereafter we refer to MPI-ESM1.2 as MPI-ESM", but then on page 11, line 296 you say "SOCOLv4 is based on the latest version of MPI-ESM (v1.2.01p6)". I'd suggest removing the version number here and putting it in the earlier section.

Corrected

Figure 3b: given SOCOLv4 uses an interactive ocean, is a time-series comparison of Nino3.4 index the best method of comparing these quantities? Perhaps histograms similar to Figure 1 from Nowack et al. 2017 (https://doi.org/10.1002/2016GL072418) would be a better way to present these data?

We have additionally performed the Fast Fourier Transform (FFT) calculations to illustrate the power spectrum and a PDF as in Nowack et al. (2007) and added those as an extra figure to the main text (and removed panel b from Figure 3 in the manuscript):

[Figure]

Figure 1. (a): 1980-2018 Nino3.4 power spectra (°C/cycles mo$^{-1}$) calculated from the ERA5 data (orange) as well as from the three SOCOLv4 ensemble members (black). Dashed lines 95% significance levels based on an auto-regressive AR(1) process fitting. (b): histogram of Nino3.4 temperature anomalies.

Performed analyses indicate that a 40-year period is not long enough for a robust estimation of ENSO frequencies, which is expressed as a pronounced variability among the ensemble members. However, the common feature they share is that ENSO events in the model mostly last longer than in observations (the power spectrum is shifted to shorter frequencies and longer periods). Our FFT results are very similar to the MPI-ESM results presented in the shown here Figure 13 of Müller et al. (2018), where they analysed a 140-year period:

**(a) MPI-ESM-LR**

**Figure 13.** Nino3.4 power spectrum for (a) piControl of MPI-ESM1.2-LR and (b) piControl of MPI-ESM1.2-HR. Thin black lines in upper panels shows spectra of HadISST1.1 for the period 1870–2010 and thick black lines the spectra of the models. Also shown are the 95% significance levels based on an auto-regressive AR(1)-process for (blue) HadISST1.1 and (red) MPI-ESM. The $y$ axis denote the power, and the $x$ axis denote frequency (cycle/month) and years. (c) Standard

We can conclude that SOCOLv4 behaves very similar to its base model MPI-ESM in terms of ENSO periodicity and amplitude. We have now corrected this paragraph in the text, including a more detailed discussion of the biases and the reference to Müller et al. (2018).

Page 14, line 381: "The left panel of Fig. 4" - I would suggest ensuring that all figures have each sub-figure labelled (e.g. a, b, c etc.) and specific sub-plots or a range of sub-plots are referred to using these, e.g "Figures 4a - 4c..." etc. Some figures don't label their sub-plots (7, 8, 9, 12, A2), whereas other figures do (3, 4, 5, 6, 10, 11, 13, 14, 15, 16, 17, A1, A3). I would advise labelling all for consistency and clarity, and avoid using references similar to "the first plot in the left column" etc.

Corrected everywhere

Figures 4, 5, 6, 7, 8, 9, 10, 11, 12, 13, 14, A1, A2, & A3 do not have a colour-bar despite having coloured contours. While the black contours are labelled, I suggest also including the colourbar on these plots.

Most of the figures are multi-panel with different panels having different data ranges, so each figure would require several colourbars. This would occupy some space and make the instructive parts smaller. We decided to keep the figures without colourbars but reworked some of them in terms of the amount of coloured and labelled contours to make them clearer and more informative. We think that labelled contours are easier to follow, and colours are supplementary, especially for the colour-blind people like the first author of our paper.

Page 24, line 523 - what is being overestimated at the entry point? Is it the water vapour concentration itself, or something else?

We meant the overestimation of $H_2O$ at the entry point. We clarified this in the text now.

Page 27, line 586: "$CLONO_2$" rather than $ClONO_2$

Corrected

In Figure 13 I would be interested in seeing difference plots of total ozone, especially due to the stated reduction in TOC in SOCOLv4 coming from the corrected photolysis rates (page 31, lines 675-676).

We have added a figure with differences to the appendix and adjusted the related main text.

[Figure]

Figure 2. Mean seasonal cycle differences between the total ozone simulated by SOCOLv4 and SOCOLv3 and estimated by MSRv2 reanalysis and SBUV observations.

The image quality of Figure 14 seems lower than the other figures, is it a lower quality raster image?

Yes, it was just an unlucky rasterization. All figures are available in the vector format.

This is just a comment, but I found the differences in TOC coming from the aerosol climatology used for CCMI used and seen in Figure 15 very interesting.

Yes, it is an interesting feature, which also speaks in favour of interactive models, since the prescribed observation-based fields are not necessarily 100% correct.

Figure A2 caption: "ClNO2" rather than ClONO2.

Corrected

Page 43, line 879: there is a gap in the DOI for the SOCOLv4 source code link

Corrected

I would advise (although I'm sure you will do this in any case) to double-check references to pre-prints and discussion papers. For instance the Keeble et al. (2020) reference is now published, which occured after the discussion started for this paper.

All checked and updated

**Response to comments by Referee 2**

As a major novelty in comparison to the predecessor SOCOLv3 it is mentioned that the model is now coupled interactively to an ocean and interactive aerosol at the same time. It would be good to say clearly if the authors think that any of the evaluated features are influenced by this novelty. I wouldn't consider it a problem if not but also if that is the case it should be clearly spelled out.

The interactive ocean in the new SOCOL version is an advantage but rather in the context that it broadens the range of potential model applications, since now the model can be easily applied for the future and paleo studies, as well as to any other topics that involve large perturbations to the system like major volcanic eruptions, anthropogenic activity, geoengineering, etc. However, the interactive ocean model also brings its biases, as it is imperfect compared to reality like any other model, therefore the SOCOLv4 run with historically prescribed ocean could well be better than the coupled version in certain aspects of climatology. We don't analyse such effects in the current paper, however such runs (interactive and prescribed ocean for the same period) are currently being performed within the framework of CCMI2 activity (http://blogs.reading.ac.uk/ccmi/ccmi-2022/).

In this context I'm a bit at odds with the statement in the introduction that "the interaction of Earth system components is required for reasonable model performance in most cases". What are these cases? And if it really were "most cases", it should be possible to clearly identify the benefits of the full coupling in the model evaluation.

We describe different cases in further paragraphs in the introduction, namely highlighting the roles of interactive chemistry and resolved middle atmosphere and mentioning the evolution of CMIPs. However, we agree that the statement "*most cases*" is probably too strong, as the cases are difficult to quantify and different scientific questions require different modelling setups and full interactivity in certain cases can even be excessive, so we changed it to "***many** cases*".

Model tuning: A major component of climate model development is the model tuning which should be adequately represented in the model description. My understanding is that the authors haven't changed anything in the model tuning in comparison to the reference model (without chemistry and aerosols) as presented by Mauritsen et al. (2019). If that is true it should be mentioned explicitly, if not, the tuning strategy would need to be described.

We didn't change anything in the tuning of the base model. We now added a sentence to chapter 2: "*Although other higher horizontal and vertical resolutions of MPI-ESM are also tuned and available for use, we chose the LR configuration since it is the most used, better tuned (Mauritsen et al., 2019), and better suited for long-term climate simulations in terms of required computational resources and storage. **It must be noted that we didn't change anything in the tuning of the MPI-ESM1.2 LR model version described in Mauritsen et al. (2019). All our changes refer to the coupled chemistry and sulfate aerosols modules.***"

Computational performance: I would find some information useful on the increase of computing time due to the interactive treatment of chemistry and aerosols in comparison to the base model. I know that this may be machine and configuration dependent but some numbers for typical configurations would be informative.

We have added a sentence to chapter 2: "***Mostly due to the large number of new tracers introduced, SOCOLv4 is about 2.6 times slower than MPI-ESM, 30% of which is from AER.***"

Figures: Most of the figures appear fairly blurred in the pdf accessible to me. This should be taken care of before publication.

All figures will be provided in the vector format

L95: "an upgraded dynamical core" Has there really been a change of the dynamical core in the family of ECHAM atmosphere GCMs since version 4? If yes, please specify.

"*Dynamical core*" is changed to "***atmospheric model***".

L153 My understanding is that the sea ice model has plenty more than just two parameters that could be used for tuning. But it is also not clear to me what the authors mean when they speak of "parameterizations of changes in ice". As said above, if the authors did any tuning it should be mentioned.

Changed to: "***The calculations of sea ice concentration and thickness are based on Semtner (1976) formulation and tuned in MPI-ESM to produce the annual average preindustrial Arctic sea ice volume of roughly 20–25 thousands of km³ (see Mauritsen et al., 2019).***"

L193 "tropospheric GW sources prescribed as a function of latitude": Which GWs? Was this option used in these experiments? If yes, using what function.

As there are actually more changes in ECHAM6 GW schemes compared to ECHAM5, we have corrected this part as: "***As in ECHAM5***, *gravity waves drag (GWD) is calculated using a subgrid orography scheme (Lott, 1999). The propagation and dissipation of the waves follows the formulation of Palmer et al. (1986) and Miller et al. (1989). Non-orographic GWD parameterizations is based on a wave-spectrum approach (Hines, 1997a, b).* ***However, some parameters of both orographic and nonorographic gravity wave schemes have been adjusted for use in ECHAM6 during the tuning process of MPI-ESM (see Mauritsen et al., 2019).*** "

L194 "RRTMG": If SOCOL follows Mauritsen et al. (2019), then the PSRAD variant of RRTMG is used, which should be mentioned.

Corrected as: "*Both long-wave and short-wave radiative transfer calculations are now described by the* ***PSrad scheme (Pincus and Stevens, 2013), which is based on the*** *k-correlated method of RRTM-G (Iacono et al., 2008).*"

L198 "radiation calculation of SCOCOLv4": I guess this is true only for the radiation calculation interacting with the model dynamics, not photochemistry, right? It would be useful to mention that these things are, to my understanding treated inconsistently as it is done in many other CCMs.

Yes, we have a separate treatment of radiation for dynamics and photochemistry. We have added a sentence in this paragraph: "***Note that for photolysis rates calculations, we use a separate subroutine based on the application of look-up-tables (Rozanov et al., 1999).***"

L198 "uses prognostic tracer concentrations … except CO2" This seems to clash with L305 where it is claimed that concentrations of GHGs from scenarios are used. Or do you mean emissions for the latter? What about CH4 and N2O, which are not mentioned at line 305?

We use concentrations for all GHGs, but for CO2 it is applied globally, while for others only at the lowermost model level. We have added a sentence to make it clear: "**Concentrations for CH4, N2O, and CFCs are prescribed at the lowermost model level, while CO2 is prescribed for the entire atmosphere .**"

L202 MACv2-SP: I thought that SOCOL calculates at least sulfate aerosols interactively. Please explain where the climatology is used and where interactive aerosol.

AER sulfate aerosols are used only in the stratosphere. In the troposphere, we have two sources of tropospheric aerosols, which also include sulfate: MACv2-SP (optical properties) for radiation and GADS (ND and SAD) for chemistry. We have corrected several parts throughout the text to make this clearer.

3.1.2: In particular here it would be interesting to know if SOCOL performs in any way different than the base model without interactive chemistry and aerosols.

We do cover this question briefly in the end of this subchapter and refer to Figure A1 in the appendix, where we show that most of the biases in SOCOL4 climatology of temperature and winds are inherited from MPI-ESM, but there are also now changes in the middle atmosphere related to QBO-nudging and different ozone.

ENSO evaluation: I don't think a Nino3.4 time series is very useful for evaluating the model performance. A power spectrum and teleconnection patterns would be much more helpful. But again, if SOCOL performs very similar to the base model it would be sufficient to just state that.

Please see the response to Referee 1 (Figure 1a,b). Overall, we have additionally performed and FFT and PDF analyses of the ENSO data, which revealed results that are very similar to the base model. We have corrected the related text in the paper, making it more detailed about the biases in the ENSO frequency.

L398 "conclude that forcing and model response are adequately represented": Well in principle it could also be that the forcing is too strong and the response too weak.

We have removed this sentence, as the conclusion on the warming amplitude and pattern are already presented before.

L422 It would be good to explain the QBO nudging approach or to provide a reference.

We have added "**(see chapter 2.4)**", where it is described.

L615: For the ozone bias in the lowermost stratosphere only chemistry is discussed as a potential reason. Could it be that the tropopause is at a slightly wrong altitude which may have large effects in this regions of large vertical gradients?

True, especially that we do have a downward displacement of the extratropical tropopause, which we mentioned in chapter 3.1.2. In our additional sensitivity tests, we performed nudged SOCOLv4 simulations and identified that this displacement can introduce ozone high biases when compared to observations, because then stratospheric air is compared to tropospheric ozone values. This issue is mostly important in the lowermost stratospheric altitudes that are statistically insignificant in Fig. 11, but we still added a sentence there highlighting the potential role of dynamical factors: "***The lower stratosphere is a complex region with competing transport, mixing, and chemistry processes, variations of the tropopause, and exchange with the troposphere (Holton et al., 1995), which makes it difficult to identify exact factors responsible for these biases.***". We've also adjusted the rest of the text there to fit the context.

 L656 "southern(polar) night jet"

corrected

Figure in the appendix: I don't know the journal policy concerning this, but if the appendix contains only three figures, I would consider it more convenient to not have an appendix but include the figures in the main part.

We've put these figures to the appendix as their role was rather supportive for the general discussion, and the related parts in the text are not long, so adding them to the main part can make the paper too busy with figures. We are, however, open to hear the opinions of the editor and typesetters.

Figure 15: I wouldn't speak of "normalized", but that an anomaly with respect to some reference is presented.

Right. Corrected to "***anomalies with respect to the mean over the whole period***".

L769 "whereas for SOCOlv4": I find the whereas confusing because at the first read I related it to the tropopause definition.

Changed to "***while***"

L807 "underestimating the aerosol lifetime" Well, if the observations are flawed, as mentioned in the following sentence, it's not sure that this is an underestimation.

Corrected as "***potentially*** *underestimating*" and adjusted the following text. We still consider this bias as potentially systematic, since it is pronounced for all three large events, while the instruments were changing (even though all of them were having their own problems).

L822 "SOCOL includes an interactive ocean" As said in the beginning: If this is put so prominently as an achievement, effects of that should be discussed.

Modified as "*Unlike its predecessor chemistry-climate model SOCOLv3, SOCOLv4 includes **both** the interactive ocean and the sulfate aerosol module, **which significantly broadens the range of potential model applications.***"

**References**:

Holton JR, Haynes PH, McIntyre MI, Douglass AR, Rood RB, Pfister L. Stratosphere-Troposphere Exchange, Review of Geophysics, 33, 4,  pp. 405. 1995.

Müller, W.A., Jungclaus, J.H., Mauritsen, T., Baehr, J., Bittner, M., Budich, R., et al. (2018). A higher-resolution version of theMax Planck Institute Earth System Model (MPI-ESM1.2-HR). Journal of Advances in Modeling Earth Systems,10,1383–1413. https://doi.org/10.1029/2017MS001217

Nowack, P. J., P. Braesicke, N. Luke Abraham, and J. A. Pyle (2017), On the role of ozone feedback in the ENSO amplitude response under global warming, Geophys. Res. Lett., 44, 3858–3866, doi:10.1002/2 016GL072418

Pincus, R., & Stevens, B. (2013). Paths to accuracy for radiation parameterizations in atmospheric models. Journal of Advances in Modeling Earth Systems, 5, 225–233. https://doi.org/10.1002/jame.20027

Semtner, A. J. (1976). A model for the thermodynamic growth of sea ice in numerical investigations of climate. Journal of Physical Oceanography, 6, 379–389.